# How modelling paradigms affect simulated future land-use change

Calum Brown[1], Ian Holman[2], Mark Rounsevell[1,3]

[1] Institute of Meteorology and Climate Research, Atmospheric Environmental Research (IMK-IFU), Karlsruhe Institute of Technology, Kreuzeckbahnstraße 19, 82467 Garmisch-Partenkirchen, Germany
[2]School of Water, Energy and Environment, Cranfield University, Vincent Building, Bedford MK43 0AL, UK
[3]School of Geosciences, University of Edinburgh, Edinburgh EH8 9XP, UK

*Correspondence to*: Calum Brown (calum.brown@kit.edu)

**Abstract.** Land use models operating at regional to global scales are almost exclusively based on the single paradigm of
economic optimisation. Models based on different paradigms are known to produce very different results, but these are not always equivalent or attributable to particular assumptions. In this study, we compare two pan-European integrated land use models that utilise the same climatic and socio-economic scenarios, but which adopt fundamentally different modelling paradigms. One of these is a constrained optimising economic-equilibrium model and the other is a stochastic agent-based model. We run both models for a range of scenario combinations and compare their projections of spatially aggregate and
disaggregate land use changes and ecosystem services supply levels in food, forest and associated environmental systems. We find that the models produce very different results in some scenarios, with simulated food production varying by up to half of total demand, and the extent of intensive agriculture varying by up to 25% of the EU land area. The agent-based model projects more multifunctional and heterogeneous landscapes in most scenarios, providing a wider range of ecosystem services at landscape scales, as agents make individual, time-dependent decisions that reflect economic and non-economic motivations.
This tendency also results in food shortages under certain scenario conditions. The optimisation model, in contrast, maintains food supply through intensification of agricultural production in the most profitable areas, sometimes at the expense of land abandonment in large parts of Europe. We relate the principal differences observed to underlying model assumptions, and hypothesise that optimisation may be appropriate in scenarios that allow for coherent political and economic control of land systems, but not in scenarios where economic and other scenario conditions prevent the changes in prices and responses
required to approach economic equilibrium. In these circumstances, agent-based modelling allows explicit consideration of behavioural processes, but in doing so provides a highly flexible account of land system development that is harder to link to underlying assumptions. We suggest that structured comparisons of parallel, transparent but paradigmatically distinct models are an important method for better understanding the potential scope and uncertainties of future land use change, particularly given the substantive differences that currently exist in the outcomes of such models.

## 1 Introduction

Computational models of the land system make essential contributions to the exploration of environmental and socio-economic changes, supporting efforts to limit climate change and reverse biodiversity loss (Harrison et al. 2018; Rogelj et al. 2018). Such models are particularly useful for exploring conditions that do not currently exist and cannot therefore be observed, as well as for understanding past and present land use impacts (Filatova et al. 2016; IPBES 2018; Smith et al. 2019). As a result, the scope and complexity of land system models have been steadily increasing, with many now representing multiple land sectors (e.g. agriculture, forestry and urbanisation) within an Earth System context (e.g. incorporating economic, climatic, hydrological and energy systems) (Harrison et al. 2016; Kling et al. 2017; Pongratz et al. 2018).

Nevertheless, simulating expected or desired future changes under novel circumstances remains a substantial challenge. Because comparable, alternative findings are rare, model results often go unchallenged, and may be misinterpreted as predictions of how the future will develop rather than projections dependent upon underlying assumptions (Low and Schäfer 2020). This could be particularly misleading in social systems such as those underpinning human land use, where no universal laws or predictable patterns exist to guide the representation of human behaviour in models. Modellers must therefore choose between a range of contested theoretical foundations, practical designs and evaluation strategies (Brown et al. 2016; Meyfroidt et al. 2018; Verburg et al. 2019).

In this complex context, the proper analysis and interpretation of model outputs is just as important as proper model design, but has received less attention. Steps such as standardised model descriptions, open access to model code, robust calibration, evaluation and verification, benchmarking, uncertainty and sensitivity analyses are all necessary to ensure that model results are interpreted appropriately (Baldos and Hertel 2013; Sohl and Claggett 2013). Currently, few if any of these steps are taken universally and rigorously in land use science (van Vliet et al. 2016; Brown et al. 2017; Saltelli et al. 2019). This study focuses on one in particular; the comparison or benchmarking of independent land use models against one another.

Comparison is especially important for land use models because a range of very different conceptual and technical approaches could be valid for simulating social-ecological dynamics (Filatova et al. 2013; Brown et al. 2016; Elsawah et al. 2020). In the absence of fair comparisons, it is impossible to objectively choose between these approaches or to identify the assumptions on which their outputs are most conditional. However, while comparisons of model outputs have been made (Lawrence et al. 2016; Prestele et al. 2016; Alexander et al. 2017), their ability to link particular outputs to particular methodological choices has been limited by the sheer number of differences between individual models. Alexander et al (2017), for instance, found that model type explained more variance in model results than did the climatic and socio-economic scenarios, but they were not able to determine exactly why.

These previous comparisons reveal a major challenge: the shortage of models that take distinct approaches in similar geographical and thematic areas, and which would therefore allow for more controlled and informative comparison exercises. Most established models, especially those operating over large geographical extents, share a basic approach that optimises land use against economic, climatic and/or environmental objectives. Technical and geophysical constraints are often treated

in detail, while social, institutional and ecological factors are rarely included (Brown et al. 2017; de Coninck et al. 2018;
Obermeister 2019). Large areas of system behaviour remain under-explored as a result (Brown et al. 2016; Huber et al. 2018; Meyfroidt et al. 2018), with the likely consequence that established findings have implicit biases and blind spots. These can be especially problematic for the simulation of future scenarios in which neglected aspects of land system change become prominent (Estoque et al. 2020), and can be partially if not fully revealed by structured comparison exercises.

In this article, we take advantage of the development of two conceptually distinct, but practically equivalent models of the
European land system to make a direct comparison between alternative modelling paradigms. We use the term 'modelling paradigm' here to refer to a methodological approach that is based on a distinct theoretical description of the system in question; in this case 'top-down' and 'bottom-up' approaches frequently identified as paradigms in the literature (Brown, Brown, & Rounsevell, 2016; Couclelis, 2002). These models, an Integrated Assessment Platform (IAP) and an agent-based model (ABM) share input data to run under the same internally consistent scenario combinations. The former is a constrained optimising
economic-equilibrium model and the latter is a stochastic behavioural model. We run both models for combinations of the Representative Concentration Pathways (RCP) climate scenarios and Shared Socioeconomic Pathways (SSP) socio-economic scenarios (O'Neill et al. 2017), and compare their projections of territorial and aggregate land use change and ecosystem service provision. We use this analysis to understand the effects and importance of the different assumptions contained in each model for simulated land use futures, and draw general conclusions about the contributions of both approaches to
understanding land system change.

## 2. Methods

This paper uses two contrasting models of the European land system: CRAFTY-EU (Brown, Seo, & Rounsevell, 2019) and the IMPRESSIONS Integrated Assessment Platform (IAP) (P. A. Harrison, Holman, & Berry, 2015; Paula A. Harrison et al., 2019). Both models cover all European Union Member States except Croatia, as well as the UK, Norway and Switzerland.
The IAP's simulated baseline land use map, land use productivities, scenario conditions and ecosystem service provision levels were used in CRAFTY-EU, making them uniquely equivalent examples of different modelling paradigms (Fig. 1). Both models were run for a subset of socio-economic and climatic scenario combinations, and their outputs systematically compared, as described below.

### 2.1 Model descriptions

The *IMPRESSIONS IAP* is an online model of European land system change that incorporates sub-models of urban development, water resources, flooding, coasts, agriculture, forests and biodiversity. Within this cross-sectoral modelling chain, rural land use is allocated within 30-year timeslices according to a constrained optimisation algorithm that maintains equilibrium between the supply and demand for food and (as a secondary objective) timber, through iterating agricultural commodity prices (cereals, oilseeds, vegetable protein, milk, meat etc.) to promote agricultural expansion or contraction
(Audsley et al., 2015). This model therefore aims to satisfy food demand (taking account of net imports), and does so optimally

subject to constraints imposed by biophysical and socio-economic conditions. Calculations are carried out across overlapping geographically unstructured clusters of cells with similar biophysical conditions (based on soil and agroclimate), with profitability thresholds used to determine which land use and management intensity offer the greatest returns across each cluster. Land use proportions within each 10' x 10' grid cell represent the aggregations of the optimal solutions for each (up to 40) associated cluster. At cell level, this aggregation therefore represents the (spatially weighted) optimised land use solution for each cluster containing the grid cell in question. The clustering recognises that different biophysical conditions (soil and agroclimate) differentially influence the suitability, productivity and profitability of different crops and different agricultural systems (arable, dairy etc.), leading to heterogeneity in agricultural land use within a grid cell. The IAP runs from a present-day simulated baseline land use configuration to the mid-2080s under combined climatic and socio-economic scenarios. The IAP has been applied and evaluated in a large number of studies including sensitivity and uncertainty analyses (e.g. Brown et al. 2014; Harrison et al. 2015, 2016, 2019; Kebede et al. 2015; Holman et al. 2017a, b; Fronzek et al. 2019). A full model description and the online model itself are available at http://www.impressions-project.eu/show/IAP2_14855.

*CRAFTY-EU* is an application of the *CRAFTY* framework for agent-based modelling of land use change (Brown, Seo, et al., 2019; Murray-Rust et al., 2014) that covers the same extent as the IAP at the same (10 arcminute) resolution. CRAFTY uses the concept of Agent Functional Types (AFTs) (Arneth, Brown, & Rounsevell, 2014) to simulate land use change over large geographical extents while capturing key behaviours of decision-making entities (agents) that include individual land managers, groups of land managers and institutions or policy bodies (Holzhauer, Brown, & Rounsevell, 2019). Modelled land manager agents compete for land on the basis of their abilities to produce a range of ecosystem services that society is assumed to require. In CRAFTY-EU, these services include provisioning (food crops and meat, timber), regulating (carbon sequestration), cultural (recreation) and supporting services (habitat provision through landscape diversity) . The abilities of agents to supply these services under given biophysical and socio-economic conditions are derived either from IAP model results (Fig. 1) or from basic assumptions linking land uses to service levels, as explained in Brown et al. (2019). Satisfying demands for services brings economic and non-economic benefits to individual agents, with benefits quantified as functions of unsatisfied demand. In this case, these functions are linear and equivalent for all services, meaning that the benefit of production of each service increases equally per unit of unmet demand, providing a clear basis for model comparison. Economic benefit represents income from marketable goods and services, and non-economic benefit represents a range of motivations, from subsistence production to the maintenance of societal, cultural or personal values associated with particular services or land uses. Ecosystem services production levels are determined by the natural productivity of the land and the form and intensity of agents' land management. The outcome of the competitive process at each annual timestep is determined by agent-level decision-making that is not constrained to generate the greatest benefit, and agents are parameterised here to continue with land uses that provide some return rather than abandon their land, but to gradually adopt significantly more beneficial alternatives if available.

Importantly for this study, CRAFTY-EU is parameterised on the basis of the IAP, taking IAP outputs as exogenous conditions and replacing only the land allocation component to provide alternative land use projections under identical driving conditions

(Fig. 1). CRAFTY-EU is initialised on the IAP's baseline map, and is known to only diverge from that stable baseline 'solution' as scenario conditions change (Brown, Seo, et al., 2019). Land use productivities, in terms of potential yields and ecosystem service provision levels of the simulated land use systems under the agronomic scenario conditions at cell scale, are also calculated from IAP outputs dependent on land use allocation, with the result that productivities are set to zero where the IAP determines production to be economically infeasible. For ecosystem services with economic values (meat, crops and timber),

agents in CRAFTY therefore make production choices that conform to this basic level of economic feasibility, while still being able to select a range of economically optimal or sub-optimal land uses. A full description of the model can be found in Brown et al. (2019) and an online version with access to full model code at https://landchange.earth/CRAFTY.

## 2.2 Climate and socio-economic scenarios

Seven combinations of climatic and socio-economic scenarios were simulated, based on the Representative Concentration Pathways (RCPs) and Shared Socioeconomic Pathways (SSPs) (O'Neill et al., 2017). The RCPs and SSPs were combined taking account of internal consistency with their associated greenhouse gas emissions; RCP2.6 was combined with SSP1 and 4; RCP4.5 with SSP1, 3 and 4; and RCP8.5 with SSP3 and 5 (Table 1). The SSPs have been further developed for Europe through a stakeholder-engagement process that included interpretation and quantification of key drivers of change in land-

based sectors (Table 2a; Kok, Pedde, Gramberger, Harrison, & Holman, 2019). For this study, RCPs were simulated in the IAP using outputs from two global-regional climate models (EC_Earth/RCA4 for RCP2.6, and HADGEM2-ES/RCA4 for RCPs 4.5 and 8.5 (Table 2b; Paula A. Harrison et al., 2019)). Scenario outcomes are described for CRAFTY-EU in Brown et al. (2019b) and for the IAP in Harrison et al. (2019) and Papadimitriou et al. (2019). In addition to these established scenarios, one scenario combination (RCP4.5 – SSP3) was simulated with additional variations in model parameterisations. This scenario

was chosen as producing particularly divergent results between the two models, and parameter values were altered to constrain the differences in model responses to the scenario and so to reveal the roles of underlying assumptions in producing the observed divergence. Specifically, we increased imports in the IAP by 40% (to mimic an observed under-production of food in CRAFTY), and increased the value of food production in CRAFTY by ten times (to compensate for reductions in supporting capital levels responsible for the under-production of food).


## 2.3 Conceptual framework

The model comparison presented here is motivated by the hypothesis that the nature of simulated land use allocation is one dominant source of uncertainty in land use modelling, as opposed to uncertainty in crop yields, biophysical conditions or other land system characteristics. The selected models therefore allow us to keep the latter factors common and explore how different

factors that influence land use allocation, such as profitability, non-economic motivations, demand levels and socio-economic conditions, affect model outcomes. This is possible because the models used share much of their information and design features, but adopt distinct paradigms for modelling the process of land allocation itself (Fig. 1, Table 3).

The IAP and CRAFTY-EU belong to distinct paradigms in the sense that the IAP is an example of a 'top-down' model that simulates change at the system-level – in this case through an assumption of constrained economic optimisation - while

CRAFTY is an example of a 'bottom-up' model that simulates change at the level of individual decision-makers – in this case through an assumption of behavioural choices made at the level of local land systems (Brown et al., 2016; Couclelis, 2002). These paradigms usually have different uses and justifications: the (dominant) top-down approach is computationally efficient, tractable and more in line with economic theory, although it is rarely justified as an accurate representation of how land use decisions are made in practice (in fact the evidence tends to contradict it; e.g. Chouinard et al. 2008; Schwarze et al. 2014;

Appel and Balmann 2019). The bottom-up approach, in contrast, is more exploratory and often criticised for producing uncertain results, but explicitly attempts to achieve greater process accuracy (Brown et al., 2016).

Neither of these models is intended to accurately predict real-world land use change, but to project land system dynamics on the basis of complex and integrated processes founded on a small number of key, transparent assumptions. This comparison is therefore intended first and foremost to explore the reasons for *simulated* land use changes, and does not speak directly to

observed land use changes. Nevertheless, both models have been extensively used and evaluated, and both respond stably and predictably to driving conditions (Brown et al., 2014; Brown, Holzhauer, Metzger, Paterson, & Rounsevell, 2018; Brown, Seo, et al., 2019; Paula A. Harrison et al., 2019, 2016; I. P. Holman et al., 2017). Both also have similar uses, being intended to support academic research and education and, to some extent, capacity building with stakeholders to increase understanding of the importance of socio-economic and climatic changes, systemic inter-relationships in the land system, and geographic

regions that may be particularly vulnerable or resilient to change. As a result, the comparison does not consider model purpose or the suitability of either model for direct policy-support, prediction or other unintended uses.

Further, some of the effects of the different land allocation mechanisms contained in these models are apparent *a priori*. As a bottom-up, agent-based model, CRAFTY is less constrained than the IAP, with multiple outcomes being possible from a given set of input conditions. At the same time, land use decisions are subject to behavioural inertia in CRAFTY, with agents

unwilling to change existing land uses and motivated by non-economic factors that can counteract price signals. The IAP will always identify the optimal result subject to economic drivers and modelled constraints, and does so without reference to the previously simulated timepoint (i.e., is not path-dependent). It is therefore expected that the IAP responds to smaller changes in conditions than does CRAFTY, and that the models are likely to diverge as time goes on and as the magnitude of changes increases.

**2.4 Comparison**

In this study, both models are run until the mid-2080s (defined as a 30-year timeslice in the IAP, and the year 2086 in *CRAFTY-EU*). Both use a spatial grid of resolution 10 arcmin x 10 arcmin (approximately 16km x 16km in Europe), but simulated land classes differ between the two models (as described in Brown et al. 2019b) and are standardised here as described in Table 4, to focus on major, comparable forms of agricultural and forestry management. These aggregate land use classes are not

homogeneous or uniform across the simulations as they allow for a range of management forms within them. We therefore

also compare ecosystem service production levels, which account for actual forms of management simulated in each cell. Urban land use is not compared as its locations are shared by both models.

The comparison of these land use classes was made at two spatial resolutions: across the whole of the modelled domain (without reference to spatial configurations) and across 323 Nomenclature of Territorial Units for Statistics (NUTS2) regions. NUTS2 resolution was chosen for the spatially explicit comparison instead of the original 10' model resolution to limit the impact of relatively uninformative differences in the allocation of individual cells, and to focus instead on systematic differences in model responses to the simulated scenarios. This choice also reflects the fact that neither model is intended to predict cell-level outcomes, but to provide illustrative realisations of scenario outcomes, with the cell-level results of CRAFTY-EU differing between individual runs because the model is stochastic and path dependent. At NUTS2 level, only differences between the models affecting at least 5% of the relevant cells were included in the analysis. In the following sections (Results and Discussion), CRAFTY-EU is referred to simply as CRAFTY, for brevity.

The presentation of the results below is structured to reveal the effects of the paradigmatic differences between the models (and not to assess the models' shared characteristics). First, we compare outputs from each scenario at EU scale to identify the principal differences that arise in the simulations. Because the scenarios relate to the modelling paradigms in different ways (e.g. allowing for stronger or weaker economic signals), this allows us to link the results to particular modelling choices. We then compare results at NUTS2 level to identify relatively minor or hidden differences, before experimenting with forced convergence to test the role of particular parameters and assumptions in each model.

## 3. Results

### 3.1 EU-level aggregate comparison

The responses of the two models to scenario conditions are notably different in most cases (Figures 2 & 3, Table 1), albeit within similar broad limits (Fig. 2). The greatest similarities in terms of aggregate land use classes occur in the SSP1 simulations, where both models produce land systems that remain similar to the baseline, with large areas of intensive agriculture and small areas of land not managed for agriculture or forestry. The IAP results include more dedicated pastoral land and the CRAFTY results more forestry. In all simulations with very low climate change (RCP2.6), CRAFTY produces an under-supply of food and both models produce an under-supply of timber, and these shortfalls reduce under intermediate climate change (RCP4.5), where productivity is slightly higher (Fig. 3). CRAFTY produces smaller imbalances between food and timber supplies due to its equivalent valuation of all modelled services.

In other scenarios, the IAP responds most strongly to SSPs 4 and 5, while CRAFTY responds most strongly to SSP3. At aggregate level, CRAFTY produces similar results in the SSP4 and 5 simulations as in SSP1 (Fig. 2), though with generally less intensive agriculture and higher supply levels (even exceeding demand in the higher climatic productivities of RCP4.5 and 8.5) (Fig. 3a). In contrast, the IAP projects a dramatic move away from intensive agriculture in SSPs 4 and 5 as a

consequence of greatly increased productivity requiring a smaller agricultural area to meet demand. This loss of agricultural management in previously intensively-managed areas is far more pronounced in the IAP than in CRAFTY, where the wider

range of valued ecosystem services supports more management and, in some cases, oversupply of services (Fig. 3). The extent of agricultural abandonment is greatest in the IAP under intermediate climate change (RCP4.5), where increased yields in some areas reduce the relative competitiveness of agricultural land in less productive areas. Differences in the simulated extent of intensive arable management are equivalent to 25% of the EU land area in some cases.

SSP3 produces considerably smaller responses in the IAP, with some areas of all land use types going out of management and

with far larger areas of the intensive agriculture class remaining than in SSP4. CRAFTY outcomes for SSP3 are highly dependent on climate scenario, with RCP4.5 producing the strongest response, most notably in terms of a large shortfall in the supply of crops (of up to 56% of demand; Fig. 3a). In this case, widespread extensification of land use occurs, with little intensive agriculture remaining by the end of the simulation, and a slight increase in land going out of agricultural or forestry management. In RCP8.5 these changes are less pronounced, with only small changes from intensive agriculture to extensive

and forestry management. These changes occur because SSP3 includes deteriorating inherent agricultural productivity and also substantial declines in capital values that support land management (particularly financial, human and manufactured capitals). In CRAFTY, these simultaneous changes make it difficult for agents to maintain intensive management against competition from extensive and less capital-dependent forms of management. The increased yields in some parts of Europe produced by climate change in RCP8.5 make this scenario more conducive to the maintenance of intensive management.

The models also respond very differently to the SSP5 scenario (paired only with RCP8.5). In the IAP, large areas switch to extensive and other/no management classes while there is very little overall change in CRAFTY. The differences between the models' responses are mainly due to the higher yields and improved technological conditions in SSP5 making large areas of intensive agriculture surplus to requirements. These surplus areas are no longer intensively managed for agriculture in the IAP by the 2080s, but are in CRAFTY (resulting in over-supply of food) because they provide other services and because of the

gradual decision-making of agents that spreads abandonment decisions over multiple timesteps.

Together, these scenario results show that the IAP responds most strongly to scenarios with conditions in which agricultural productivity increases, and which therefore lead to reduced need for agricultural land and, in this model, extensification and agricultural abandonment (which occurs over larger extents in the IAP than in CRAFTY). CRAFTY responds less strongly to such conditions because agents have a (parameterizable) unwillingness to change or abandon their land use in the absence of

a more viable alternative, and because a wider range of services produce returns for those agents. Conversely, CRAFTY responds most strongly to scenarios in which conditions affecting agricultural productivity worsen because agents rely more strongly on a range of climatic and socio-economic conditions. Many of these conditions deteriorate in SSP3, making intensive agriculture less competitive than extensive agriculture or other multifunctional land uses, and causing intensive agents to be easily replaced (competition is a more rapid process than abandonment in the CRAFTY parameterisation used here).

## 3.2 Territorial comparison

Within the aggregate differences between model results exist some consistent spatial and territorial patterns (Fig. 4). Across scenarios, the IAP often places more pastoral and very extensive land use classes in western Europe in particular, while CRAFTY often has more intensive agriculture in mid-latitudes, and forest in eastern and northern areas (Figs. 4 & B1). These differences are very scenario-dependent, however, and as with the aggregate summaries above, the spatial patterns produced by one model in SSP3 resemble those produced by the other model in SSP4. In SSP4, the IAP projects substantially more very extensive agricultural management and forest management than CRAFTY, while the near-inverse is true for SSP3 (reflecting implicit assumptions that over-production is not penalised in CRAFTY, and that intensive agriculture retains an efficiency advantage over extensive in the IAP). CRAFTY also produces a great deal more forest management in RCP2.6-SSP1, with intensive arable agriculture dominating only in the most productive parts of France, Germany and the UK. SSP1 is also the scenario in which the IAP produces the most concentrated areas of intensive pastoral agriculture, particularly in Ireland, the UK and France.

Notwithstanding the smaller-scale fragmentation of land uses in CRAFTY (see below), these results show that at this aggregate level, CRAFTY has a tendency (except in SSP3) to concentrate intensive agriculture in mid-latitudes, extensive agriculture in the southern Baltic states, and very extensive land uses at the European latitudinal extremes. Forestry is distributed in the western UK and central-eastern states in particular. The IAP results are less consistent, but show a tendency to produce pastoral agriculture in the west and forestry more widely. Many of these differences may reflect the valuation of a wider range of services in CRAFTY, leading to a concentration of intensive management in the most productive areas where it can maintain relative competitiveness. As above, they also reflect the differences in the conditions that the models respond to, with the IAP particularly sensitive to changes in demand that do not have spatial manifestations, and CRAFTY more sensitive to capitals that are maximised in climatically suitable, but also politically stable and affluent countries.

## 3.3 Convergence experiment

The scenario combination RCP4.5-SSP3 was chosen as having particularly different results from the two models, and so used to examine the potential for convergence in model settings and results. In this scenario, CRAFTY produces a highly fragmented land system with areas of abandoned or extensively managed land scattered throughout Europe, and a substantial shortfall in food production. The IAP, in contrast, produces large contiguous agricultural areas with far more intensive management (albeit of greatly reduced productivity) and less forestry, satisfying food demands. To control for the main differences in scenario conditions in each model, we increased food imports in the IAP to produce lower production levels in the EU, as observed in the CRAFTY result, and we increased food prices in CRAFTY to produce greater support for intensive agriculture, as observed in the IAP result. In the absence of these major differences, any remaining divergence in model outputs could be attributed to other factors.

In terms of aggregate land system composition the changes in the IAP (an increase of 40% in food imports) did not lead to a result approaching the original CRAFTY results (Fig. 5). While the extent of intensive agricultural management did decrease, this led to widespread agricultural abandonment rather than additional extensive or forestry management (demand for which was already satisfied), with remaining food production being even more concentrated in certain intensively-managed parts of

Europe (particularly the East). Large parts of southern and northern Europe fell out of agricultural management, with other regions and countries being managed only for forestry. Other results (above) suggest that the IAP would have more closely resembled the CRAFTY result had there been an explicit driver for extensification, rather than simply an effective decrease in demand levels.

From the more extensively-managed and fragmented initial result produced by CRAFTY, a ten-fold increase in food prices

did come closer to the initial IAP result, although with more intensive agriculture and less land under other or no management. The distribution of land uses was strikingly different, however. Unmanaged land mainly occurred in the same areas, and concentrations of forestry overlapped to some extent, but the agricultural land in the CRAFTY result remained highly fragmented across much of Europe. In this case, CRAFTY produced sufficient food to satisfy demand.

## 4. Discussion

Understanding the contributions of different modelling paradigms to land use projections is important for two main reasons. The first reason is that almost all large- to global-scale land system models share a single paradigm (economic optimisation of land uses), raising the risk of biases in model results and resultant, unrecognised knowledge gaps (e.g. Verburg et al. 2019; Elsawah et al. 2020; Müller et al. 2020). The second reason is that different paradigms are known to produce very different outcomes, but for reasons that remain unclear (Alexander et al., 2017; Prestele et al., 2016). The focused comparison presented

here is therefore intended to identify and explain key differences between models representing major, distinct paradigms. While conclusions are inevitably limited by the breadth of the comparison, and in particular by the many characteristics that are shared between the selected models (Table 3), our results do reveal large and consistent differences that emerge from the different ways in which those models represent land system change.

The consequences of top-down and bottom-up perspectives are apparent in the form, extent, rate and patterns of land use

change as the models respond to scenario conditions. The IAP's consistent profitability thresholds within a deterministic optimising framework respond strongly to increasing yields or decreasing demands, when the model produces widespread agricultural abandonment outside the most productive land. Conversely, CRAFTY's heterogeneous competition process within a stochastic agent-based framework responds more strongly to decreases in productivity, when the model produces extensification and expansion of agriculture. This difference is also apparent in our convergence experiment, where increased

imports in the IAP lead to reduced agricultural area, ensuring efficient production where competitiveness is highest, rather than the extensification that CRAFTY produces. Increasing food prices in CRAFTY did generate aggregate land use proportions similar to those of the IAP, albeit with largely distinct spatial distributions, suggesting that agents become more 'optimal' in behaviour when greater competitive advantages are available.

To some extent these differences are traceable to the underlying mathematical structures of the models, with the IAP identifying
any change in optimal configurations and CRAFTY maintaining existing and multifunctional land uses where possible. But the results are also subject to model sensitivity and uncertainty. Previous analyses show that the IAP responds most strongly to changes in demand levels and climate-driven yields, and that their effects outweigh those of socio-economic scenarios (Brown et al., 2014; Kebede et al., 2015). CRAFTY has similar sensitivities complemented but not overwhelmed by simulated agent behaviour (Brown, Holzhauer, et al., 2018; Brown, Seo, et al., 2019). Together these suggest that the effects and differences we find are robust and traceable to model design interacting primarily with climatic scenarios (RCPs), and with socio-economic scenarios (SSPs) to a lesser extent.

Particularly influential is the representation in CRAFTY of individual and societal desires for a range of ecosystem services, which means that extensive management practices that provide recreation, carbon sequestration or landscape diversity, for example, are adopted instead of land abandonment. This is not necessarily tied to modelling paradigm; optimisation can in principle be performed across a range of criteria, potentially accounting for many more (economically-valued) ecosystem services, although this remains conceptually and computationally challenging (Newland, Maier, Zecchin, Newman, & van Delden, 2018; Seppelt, Lautenbach, & Volk, 2013; Strauch et al., 2019). The non-optimising representation used in models such as CRAFTY is closer to the reality of how land use actually changes (Appel & Balmann, 2019; Schwarze et al., 2014), but still requires additional parameterisation and rigorous uncertainty analysis (Verburg et al., 2019). In either case, there is strong justification for including a wide range of ecosystem services, particularly those such as carbon sequestration that may gain distinct values in different future scenarios (Estoque, Ooba, Togawa, & Hijioka, 2020; Kay et al., 2019).

One consequence of simulating demand and supply of a range of ecosystem services is that the relative economic support available for food production becomes a key determinant of the balance of different land uses as agriculture, while still dominant in area, must compete with alternative management options. Models such as the IAP seek to maintain food supplies, even at the expense of other services such as timber production, while models such as CRAFTY allow supply levels to emerge from simulated decisions and so are capable of producing shortfalls. All the results of the models are affected by this basic assumption about whether equilibrium does or will exist in the food system, and further by the extent of disequilibrium that is tolerated and the mechanism by which that extent is defined. For instance, food prices in CRAFTY can respond to shortfalls in production through a number of parametric functions, while the in the IAP prices are automatically adjusted within broad limits to ensure that demand and supply match. However, shortfalls in food production in CRAFTY do not lead to simulated hunger, societal unrest or migration, and food prices in the IAP may become unrealistically high in scenarios where economic and social conditions are very challenging (Hamilton et al., 2020; Pedde et al., 2019). In both models, the simulation of the European land system as distinct from the rest of the world requires implicit but shared assumptions about conditions in other regions and their relationships to Europe. Alternative assumptions would inevitably lead to different outcomes and, perhaps, greater differences between the two models' results. As conceptual alternatives, therefore, neither of these necessarily capture the true dynamics of food prices and production levels, which remains a major challenge for land system modelling (Müller et al., 2020; Pedde et al., 2019).

Beyond differences at aggregate level, another notable feature of results shown above are that CRAFTY produces far more small-scale heterogeneity in land use than does the IAP. This heterogeneity is particularly pronounced in CRAFTY's SSP3 simulations (Fig. 5) and reflects a basic modelling approach: the simulation of time-dependent decisions affecting individual cells, with agents parameterised here to abandon land only if it provides no returns, and then only gradually. This effectively precludes the system-level optimisation practised by the IAP, which does not account for individual land use decisions. Individual-level heterogeneity is, inevitably, very difficult to parameterise precisely, although participatory techniques have some promise in this respect (Elsawah, Guillaume, Filatova, Rook, & Jakeman, 2015). Conversely, (constrained) optimising models like the IAP produce results that may not replicate observed rates or spatial structures of land use change (Brown, Alexander, Arneth, Holman, & Rounsevell, 2019; Low & Schäfer, 2020; Turner, Field, Lobell, Sanchez, & Mach, 2018), but can introduce spatial dependencies as further constraints on optimisation in order to approximate spatially-mediated social processes such as imitation (Brown, Alexander, & Rounsevell, 2018; Meiyappan, Dalton, O'Neill, & Jain, 2014). Bottom-up models in general tend to be less precisely specified and so produce more variable results (or are more "skittish" as (Couclelis, 2002) put it). They are also generally less often compared against observational (or other modelled) data, and while their flexibility makes fitting-to-data notably feasible in principle, their inherent tendency to produce variable results means that the production of any one particular outcome does not have the apparent significance that it does for a more constrained model. Both models used here have been compared against 'observed' land use data to some extent, with an example application of CRAFTY compared and calibrated to MODIS land cover data (Seo, Brown, & Rounsevell, 2018) and the IAP (and hence, indirectly, CRAFTY) calibrated to match CORINE land cover and NUTS2 yields (P. A. Harrison et al., 2015).

Notwithstanding the gains to be made by better understanding the relative performance of different modelling paradigms, it is essential to recognise some hard limits. No land use model is intended or able to provide calibrated representations of all the mechanisms responsible for land use change, especially under imagined future conditions. Models of this kind are inevitably reductionist in nature and omit a large number of important factors and processes that occur in reality – particularly, in this case, those occurring at smaller spatial scales than are simulated here. Both alternatives must therefore be seen as providing realisations of assumptions that are useful in some ways but incorrect in others. Optimising models have the advantage of representing idealised conditions in that they maximise achievement of modelled criteria such as production levels, but do not necessarily reveal pathways by which those conditions can be reached in reality (Ligmann-Zielinska, Church, & Jankowski, 2008; Low & Schäfer, 2020). Process- or agent-based approaches, meanwhile, can allow exploration of the large behavioural uncertainties involved in the simulation of human systems, and can be powerful tools for stakeholder engagement and understanding (Low & Schäfer, 2020; Millington, Demeritt, & Romero-Calcerrada, 2011) – but are unlikely to perform any better at predicting system outcomes than simpler, more tightly constrained models (Salganik et al., 2020). Indeed, their primary strength may be their ability to use theory (and so to allow a choice among theories) as a guide to processes and conditions that empirical data and optimising models do not cover (Gostoli & Silverman, 2020). Both types of model represent abstracted units, managers and characteristics of land, which do not match exactly to real-world conditions as experienced and

determined by actors in the system (e.g. productivities and profits used to drive the models are not the same as those available to real-world land managers).

Fundamentally, no single modelling paradigm is 'correct', and future developments are likely to invalidate even those assumption that appear safest at the present time. The greatest value of these two approaches may therefore lie in their ability to provide alternatives. This value is realised only in the (currently rare) cases when analogous models with similar driving conditions but different underlying assumptions, such as those used here, are available for comparison (Müller et al., 2014; Polhill & Gotts, 2009; Rosa, Ahmed, & Ewers, 2014). Further benefits can be drawn from combinations of the two modelling approaches, although this usually involves an artificial choice of systems or scales at which top-down optimisation and bottom-up emergence are assumed to occur (e.g. Castella and Verburg 2007; Verburg and Overmars 2009; Houet et al. 2014). In addition, the benefits of using each type of model can be maximised (and the differences between them potentially minimised) by flexible multi-criteria optimisation on one hand and behavioural uncertainty analysis on the other (Brown, Holzhauer, et al., 2018; Fonoberova, Fonoberov, & Mezić, 2013; Ligmann-Zielinska, Kramer, Cheruvelil, & Soranno, 2014; Newland et al., 2018). Nevertheless, substantial efforts to increase both the diversity and coherence of land system modelling are likely to be necessary if these important gains are to be made.

## 5. Conclusions

In taking two particular models as representative of major modelling paradigms we can only draw tentative conclusions about the consequences of those paradigms for model outputs. Nevertheless, we find large, consistent differences between the models that are robust to known model sensitivities and directly traceable to basic assumptions. In particular, we find that the 'bottom-up' agent-based model produces more heterogenous, multifunctional land systems than the 'top-down' model, as expected. We also find that the models respond most strongly to different scenario conditions, despite both being sensitive to climatic effects on yields and socio-economic effects on demand levels. In particular, the constrained optimisation of the top-down model is able to capitalise on increases in productivity by utilising the best land, while the agent-based model is limited by inertia and path-dependency in simulated conditions. Conversely, reductions in productivity, including through socio-economic disruption, prompt widespread extensification of land management in the bottom-up model that is not replicated in the top-down model, as simulated agents diversify and rely on more varied or even non-economic benefits. Currently these two modelling paradigms are far apart in their projections of future change, with highly divergent outcomes for European land use and food supplies. This suggests huge uncertainty about the role the land system can and will play in societal challenges such as climate change and biodiversity loss, especially if impacts of large-scale events such as pandemics and political disruption are considered. However, this comparison suggests that such divergence, and hence uncertainty, rests largely on a few key features: in particular the assumed extent of non-economic decision-making, the relative importance that society places on cultural and regulating ecosystem services compared to provisioning, and the likely rate of land use change, including abandonment and intensification, as outcomes of human decisions. Our findings show the importance of communicating these

assumptions to model users, but also of identifying better-supported and more generally-accepted positions that narrow the gap between the current extremes of dominant paradigms.

## Code and data availability

The full model code and data for CRAFTY-EU are available for download and visualisation via https://landchange.earth/CRAFTY

The IAP is available for interactive online runs at http://www.impressions-project.eu/show/IAP2_14855 but model code is not available because the IAP utilises meta-models of several other stand-alone models under different ownership.

## Appendices

Appendix A: Land use class composition

## Author contribution

CB performed the analysis and drafted the manuscript; IH & MR assisted with planning, interpretation and writing.

## Competing interests

The authors declare that they have no conflict of interest

## Acknowledgements

This research was supported by the EU Seventh Framework Programme project IMPRESSIONS (grant no. 603416) and the Helmholtz Association. We acknowledge support by the KIT-Publication Fund of the Karlsruhe Institute of Technology

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

|  | SSP1 socio-economic conditions gradually improve through economic growth, stable government, high social cohesion and international cooperation | SSP3 social and economic conditions worsen, with limited and ineffective political responses | SSP4 large economic inequalities and fluctuations develop, low social cohesion, but high technological investment & environmental protection | SSP5 emphasis on social and economic development, fossil fuel exploitation and technology |
|---|---|---|---|---|
| RCP2.6 Very low climate change | IAP simulates more intensive and pastoral agriculture and very little forest. CRAFTY increases forest at the expense of intensive agriculture. Under-supply of timber (especially in the IAP) and under-supply of food (only in CRAFTY). |  | Widespread agricultural extensification and abandonment in the IAP, and more forestry, but with under-supply of timber (agriculture shifts to optimal areas). More intensive agriculture in CRAFTY, but with under-supply of food (agriculture persists in less optimal areas). |  |
| RCP4.5 Intermediate climate change | Small differences, with the IAP having a slight shift towards pastoral and very extensive agriculture, with less forest. | Limited change in the IAP but dramatic loss of intensive management in CRAFTY, along with fragmentation, temporal dynamism and supply shortfalls. | Widespread agricultural abandonment in the IAP. CRAFTY supply levels exceed demand |  |
| RCP8.5 High climate change |  | Limited change in both models, with more extensification, forest and multifunctional |  | Widespread agricultural extensification and abandonment in the IAP. Limited |

| | | production in CRAFTY. | | change in CRAFTY, with supply levels exceeding demand. |
|---|---|---|---|---|

**Table 1**: **Climatic and socio-economic scenario identities, summaries, and main findings.**



| Socio-economic scenario | SSP1 | SSP3 | SSP4 | SSP5 |
|---|---|---|---|---|
| Climate change (RCP pairing) | Very low / intermediate | Intermediate/ high | Very low / Intermediate | High |
| EU population change (% change from 2010) | 0.4 | -38 | -22 | 47 |
| Food imports (absolute % change) | -13 | -5 | 4 | 18 |
| Increase in arable land used for biofuel production (% change from 2010) | 9 | 19 | 9 | 14 |
| Land allocated to agri-environment schemes (e.g. set-aside, buffer strips, beetle banks) (%; baseline is approx. 3%) | 6 | 2 | 5 | 0 |
| Change in dietary preferences for beef and lamb (% change from 2010) | -82 | 0 | 0 | 53 |
| Change in dietary preferences for chicken and pork (% change from 2010) | -34 | 35 | 35 | 74 |
| Change in agricultural mechanisation (% change from 2010) | 133 | -35 | 133 | 133 |
| Change in agricultural yields (% change from 2010) | -19 | -35 | 89 | 89 |
| Change in irrigation efficiency (% change in water efficiency relative to 2010); -50% = water halved per unit food | -57 | 53 | -57 | -57 |
| Reducing diffuse source pollution from agriculture by reduced inputs of fertilisers and pesticides (higher value = less inputs) (absolute value relative to optimum nitrogen) | 1.9 | 0.9 | 0.9 | 0.9 |

| | | | | |
|---|---|---|---|---|
| Water savings due to behavioural change (% change from 2010) | 52 | 0 | 0 | -30 |
| Water savings due to technological change (% change from 2010) | 45 | 0 | 29 | 29 |
| GDP (% change from 2010) | 259 | 48 | 200 | 724 |
| Change in energy price (oil; % of 2010) | 162 | 350 | 267 | 75 |
| Household externalities (preferences for lived environment: 1 = Urban; 5 = Country). Baseline = 3 | 5 | 4 | 2 | 5 |
| Compact vs sprawled development (Low = Sprawl; Medium or High = Compact); Baseline = Med | High | Low | Medium | Low |
| Preference to live by the coast (Low – High); Baseline = Med | Low | Low | Med | High |

**Table 2a: Details of the socio-economic scenarios (Shared Socioeconomic Pathways, SSPs) as simulated by the IAP. Values are shown for the 2080s timeslice. Table adapted from Harrison et al. 2019.**


| Emission scenario | RCP2.6 | RCP4.5 | RCP8.5 |
|---|---|---|---|
| Climate change | Very low | Intermediate | High |
| GCM | EC_Earth | HadGEM2-ES | HadGEM2-ES |
| RCM | RCA4 | RCA4 | RCA4 |
| GCM sensitivity | Intermediate | High | High |
| European $\Delta T / \Delta Pr$ | 1.4°C / 4% | 3.0°C / 3% | 5.4°C / 5% |

**Table 2b: Details of the climate scenarios used in both models. RCP denotes Representative Concentration Pathway, GCM: General Circulation Model, RCM: Regional Climate Model. Change in temperature ($\Delta T$) and change in**
**precipitation ($\Delta Pr$) are relative to 1961-1990, and affect productivities as simulated by meta-models in the IAP, which are then fed into the alternative land use models (Fig. 1). Further details are available in Harrison et al. (2019).**

| | IAP | CRAFTY-EU | Key differences |
|---|---|---|---|
| **Modelling paradigm** | 'Top-down' model that represents land use change as single systemic response to drivers | 'Bottom-up' model that represents land use change as emergent from responses of multiple entities within the system | Entirely distinct conceptualisation of land use change within shared reductionist (modelling) approach |
| **Theoretical basis** | Consistent with positivist and classical economic theories of system-level dynamic equilibrium under exogenous pressures (Brown et al. 2016) | Consistent with methodological individualism and subjective expected utility theory of decision-making given uncertainty and non-economic motivations (Murray- | Neither model explicitly theory-driven but are consistent with opposing theoretical movements. |

| | | | |
|---|---|---|---|
| | | Rust et al. 2014; Brown et al. 2016) | |
| **Land allocation** | Optimisation to satisfy food demand, subject to constraints imposed by biophysical and socio-economic conditions | Individual agent decisions based on competition to satisfy demands for ecosystem services | Land allocation is imposed in the IAP but emergent in CRAFTY, and therefore more variable |
| **Variables considered (inputs)** | Defined in Table 2 | Potential and realised ES provision levels (derived from the IAP and dependent on the variables in Table 2) and agent abilities to produce ecosystem services, sensitivities to capital levels, willingness and time-dependent probability of abandoning their cells or relinquishing to other land uses when at a competitive disadvantage, and abilities to search for new cells to take over. | Most inputs are shared directly or indirectly, although the IAP more explicitly includes biophysical conditions and CRAFTY human behaviour |
| **Mathematical characteristics** | Produces single, optimal results (subject to constraints) at each timeslice | Stochastic and path-dependent; produces sub-optimal and variable results | The IAP is more mathematically constrained, but complexity of 'option space' makes results of both models difficult to anticipate |
| **Evaluation** | Extensively evaluated, including uncertainty analyses and comparison to independent data and other models (e.g. Brown et al. 2014; Harrison et al. 2015, 2016, 2019; Kebede et al. 2015; Holman et al. 2017a, b; Fronzek et al. 2019) | Extensively evaluated, including uncertainty analyses and comparison to independent data and other models (Alexander et al., 2017; Brown, Holzhauer, Metzger, Paterson, & Rounsevell, 2018; Brown, Murray-Rust, et al., 2014; Holzhauer, Brown, & Rounsevell, 2019; Seo, Brown, & Rounsevell, 2018) | No significant difference, noting that neither models targets accurate reproduction of observed changes |
| **Uncertainty & sensitivity** | Well-understood, with land use outcomes most sensitive to temperature, precipitation, yields and import levels (Kebede et al. 2015) | Well-understood, with land use outcomes most sensitive to yields (including climate effects), import levels and (to lesser extent) agent behaviour (Brown et al. 2018) | CRAFTY has sensitivities to behavioural parameters not present in the IAP |
| **Spatial resolution** | 10 arcminutes (approx. 16km in Europe), with up to 40 forms of land use | 10 arcminutes (approx. 16km in Europe), with continuous variation | Identical resolution for defined classes, but different forms and |

| | and management proportionally distributed within each cell | in characteristics within 17 forms of land use and management | extents of variation within those classes |
|---|---|---|---|
| **Temporal resolution** | Timeslices: Baseline, 2020s, 2050s, 2080s | Annual 2016-2086 | CRAFTY has higher temporal resolution |
| **Principal uses** | Research, education, capacity building (students and stakeholders) | Research, education | CRAFTY less used in stakeholder engagement |

**Table 3: Summary comparison of the two models used in this study across a range of characteristics, many of which stem from the distinct modelling paradigms used. Further details are provided in the text, and references cited there.**

| Land use classes for comparison | Explanation |
|---|---|
| Intensive agriculture | Intensive forms of agriculture primarily dedicated to crop production but including some grassland |
| Extensive agriculture | Extensive forms of arable and pastoral agriculture |
| Pastoral agriculture | Dedicated and primarily intensive pastoral agriculture |
| Very extensive management | Management for any service that is of the lowest intensity and leaves land in a near-natural state |
| Forestry | Active management for timber extraction and other forest services |
| Other/no management | Land that is not actively managed for agriculture or forestry, but which can have a range of natural or human-impacted land covers |

**Table 4: Land use classes used in the comparison and their composition. Derivations from the full range of CRAFTY and IAP classes are given in Table A1.**

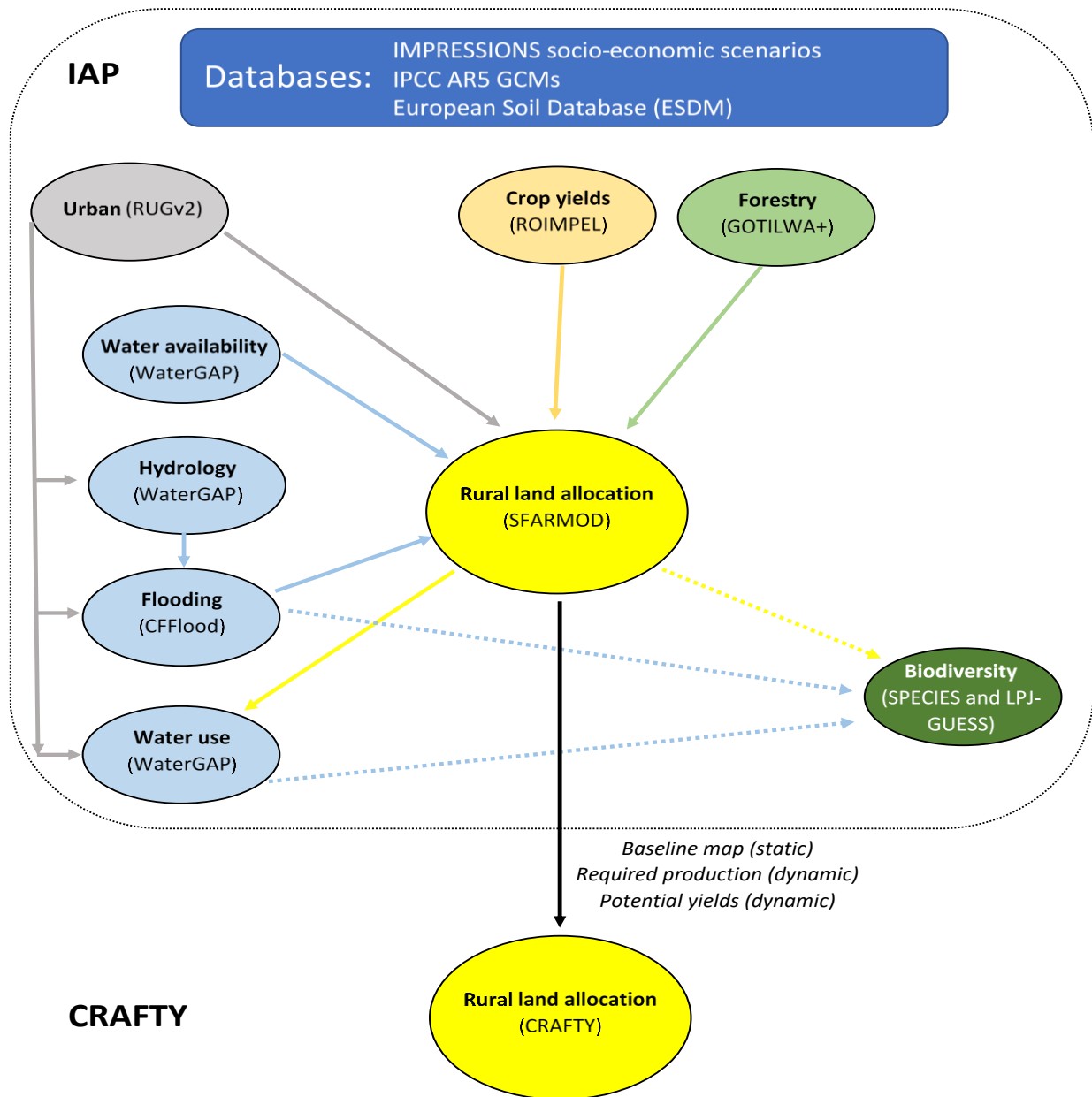

**Figure 1: Simplified schema showing the structure of the IAP in terms of its component metamodels, and its relationship to CRAFTY in this study. Results presented in this study are taken from the alternative land allocation models (yellow), and results from the biodiversity model are not used. The information transferred from the IAP to CRAFTY utilises all of the inputs to SFARMOD and describes initial and scenario-dependent conditions affecting agent decision-making in CRAFTY.**


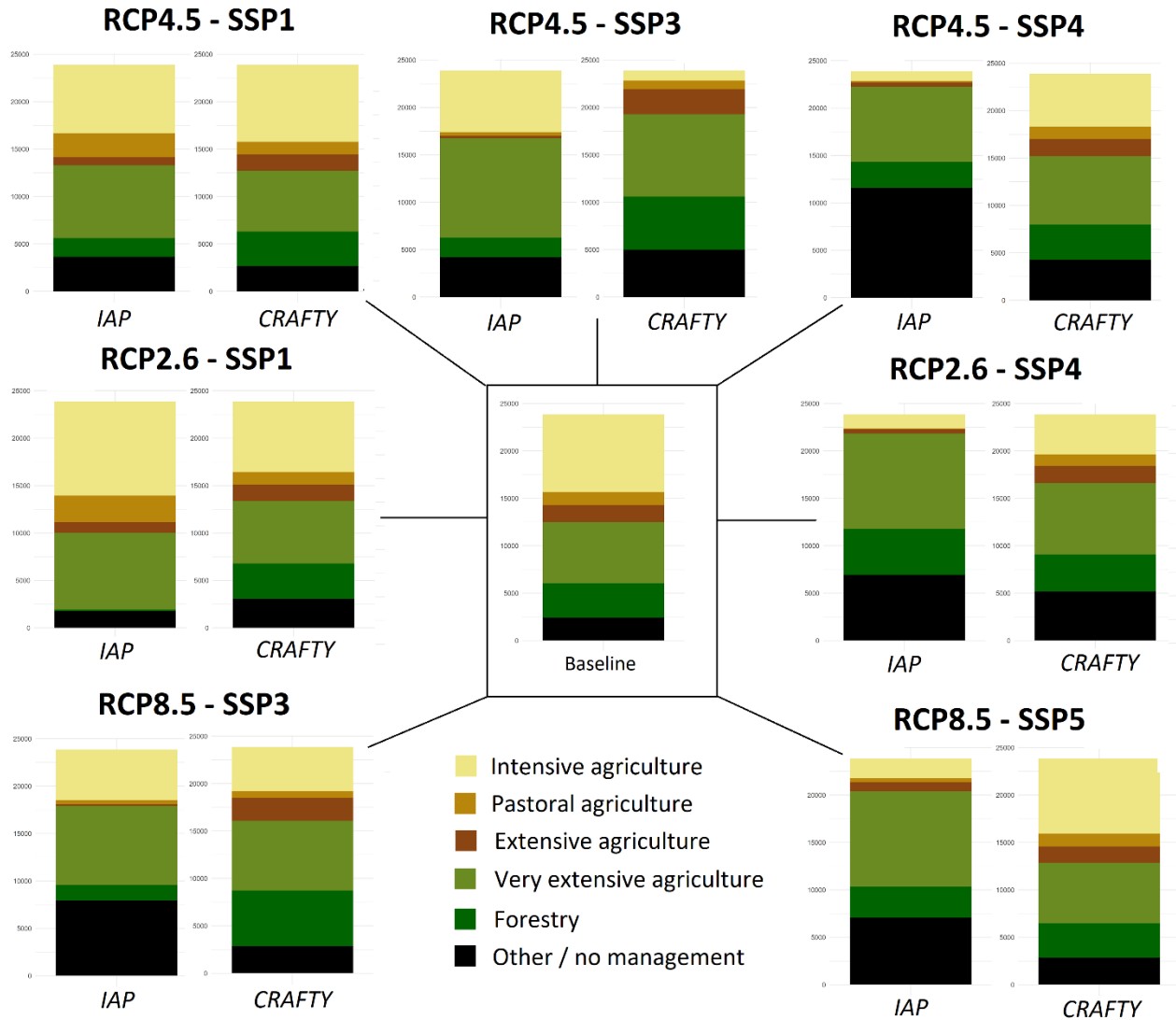

**Figure 2: Simulated land use classes for each scenario in each model in the mid-2080s. Bars show the number of cells occupied by each class, out of the total number of 23,871 cells (y-axes). Climate scenarios (RCPs) are arranged in rows. The baseline is identical in both models and so is only shown once.**

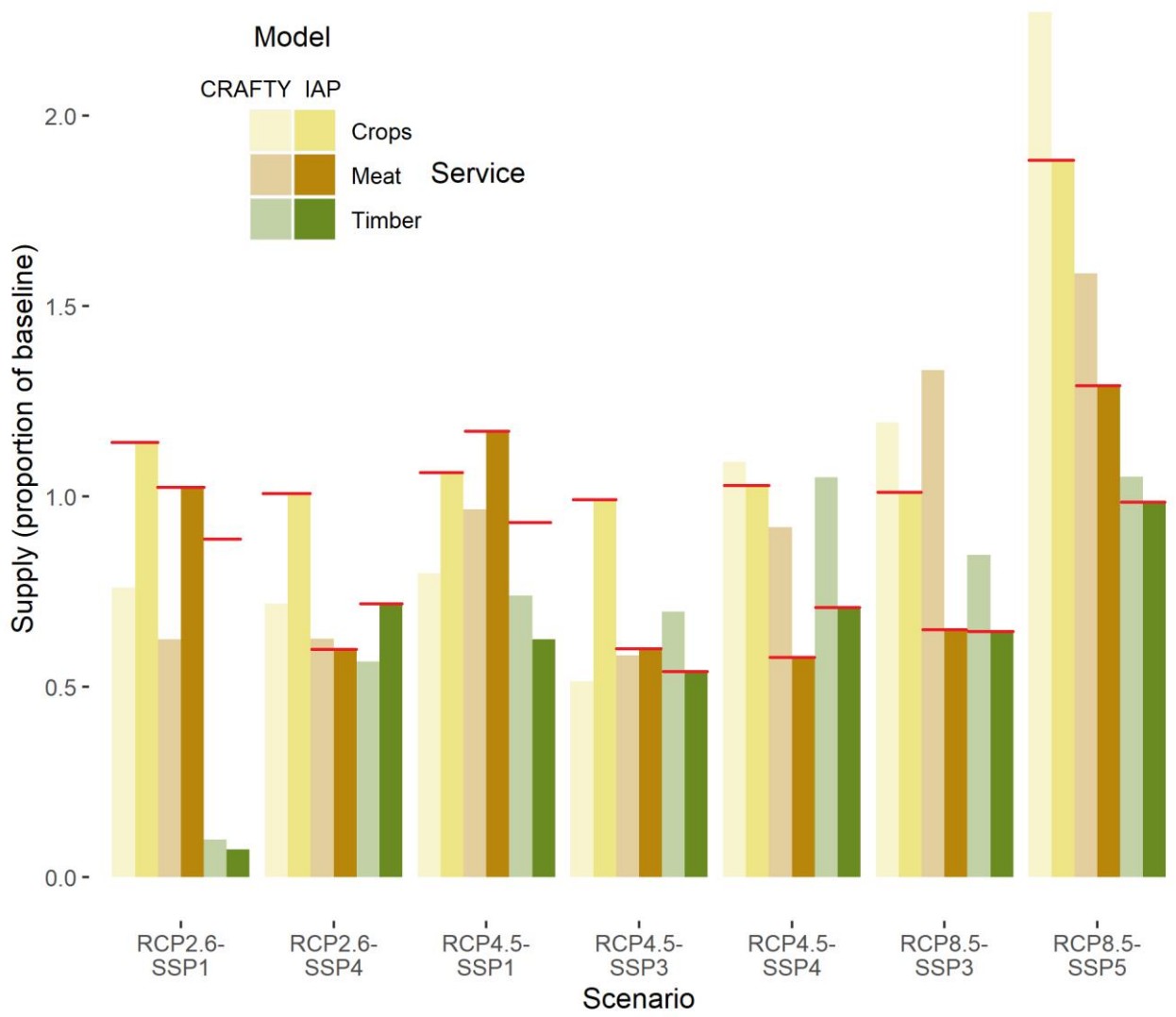

**Figure 3a: Supply levels of services that both models attempt to satisfy demand for. Supply levels are shown for each scenario, and demand levels (derived from the IAP) are indicated by a red line for each service. IAP supplies are unequal to demand levels only where the IAP reports an underproduction of a particular service (in these results, timber in SSP1 simulations). A supply value of 1.0 (y-axis) is equal to baseline supply.**

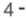

Figure 3b: Supply levels of services that only CRAFTY attempts to satisfy demands for (while the IAP does not). IAP supply levels here are calculated using CRAFTY production functions and then set as demands for CRAFTY. Demand levels are therefore equal to IAP supply by default and are not indicated by a line as in Fig. 3a. A supply value of 1.0 (y-axis) is equal to baseline supply.



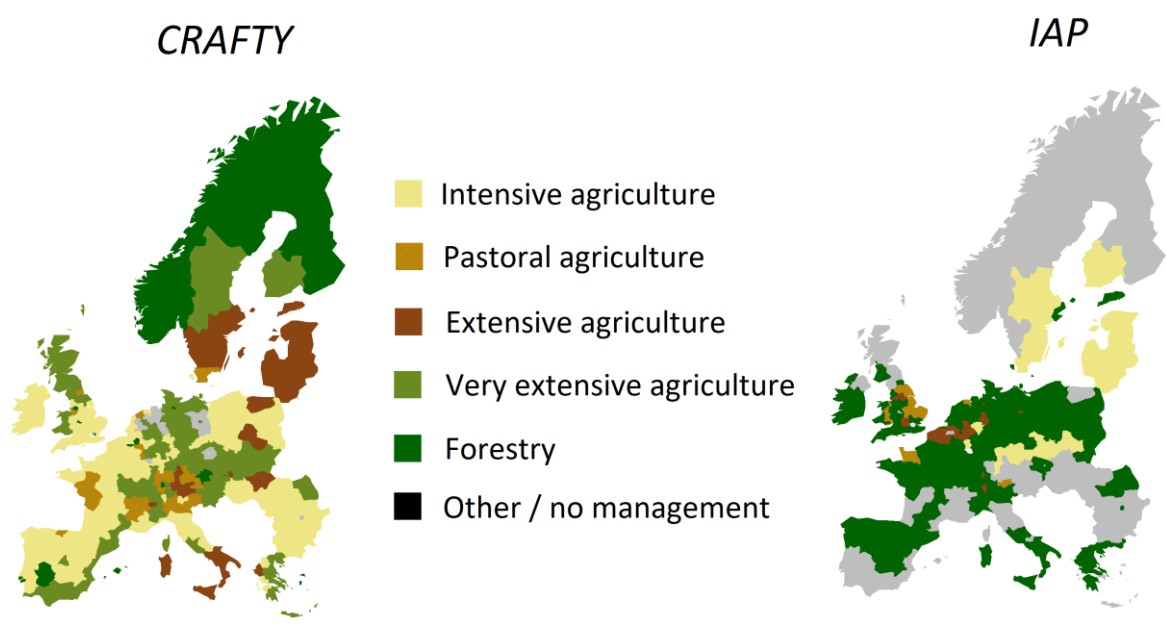


**Fig 4: Territorial differences between the models' results across all scenarios, at NUTS2 level. Colours identify the most over-represented land use type in each region in the CRAFTY and IAP results, relative to the result of the other model (i.e. the land use with the biggest difference in occurrence in that region). Grey is shown where no land use type has an over-representation of more than 5% of the region's cells. Scenario-specific results are shown in Fig. B1 (Appendix B).**


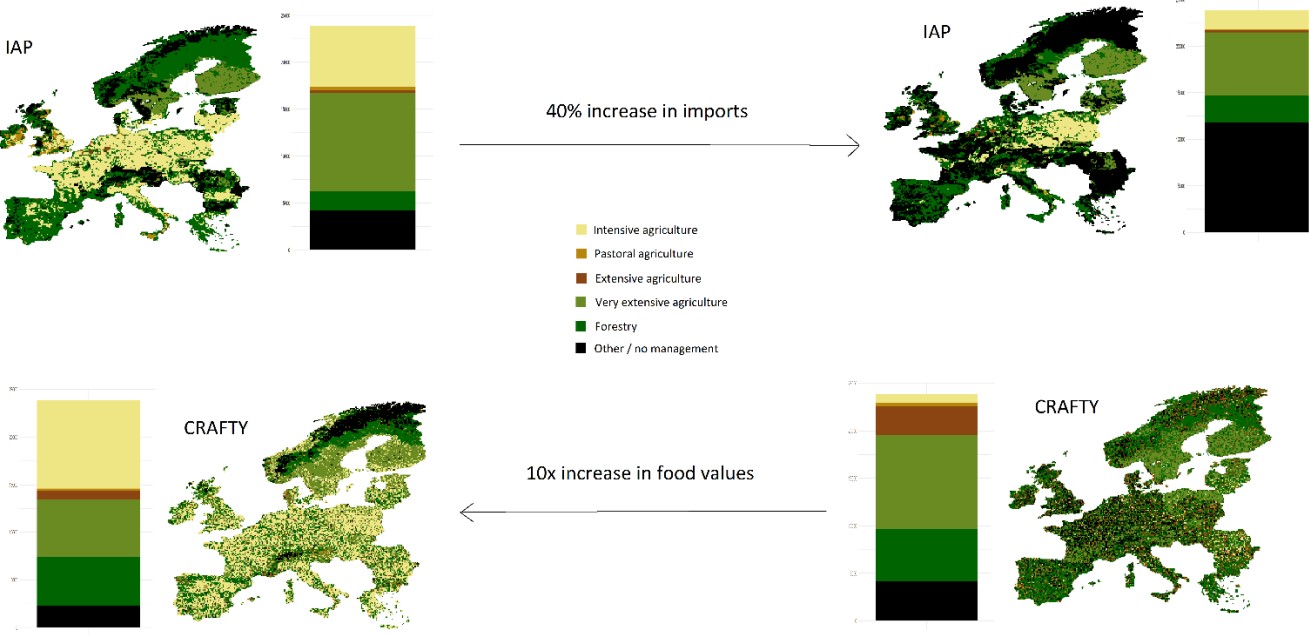

**Fig. 5: Cell-level and EU-level results for the RCP4.5-SSP3 scenario with and without alternative parameterisations designed to introduce analogous driving conditions to each model in turn. The IAP experiment is shown on the top row, and the CRAFTY experiment on the bottom. The original IAP result (top-left) moves towards the original CRAFTY result (bottom-right) with a 40% increase in imports allowing less production of food within Europe, resulting in widespread land abandonment in the new IAP result (top-right). The original CRAFTY result (bottom-right) moves towards the original IAP result (top-left) with a 10-fold increase in food prices, used to stimulate production, resulting in far more intensive agriculture in the new CRAFTY result (bottom-left). In neither case do the new results reproduce the original extremes.**

**Appendix A: Land use class composition**

Ecosystem services production in CRAFTY is derived from that of the IAP, which uses a suite of meta-models to simulate production levels as described in (Paula A. Harrison et al., 2019), and is presented in detail in Brown et al. (2019). CRAFTY-EU also shares a baseline map with the IAP, with the aggregated land use classes used here derived from CRAFTY's Agent Functional Types (AFTs) and the IAP's land use classes as described in Table A1.


| Agent Functional Type | IAP Class | Aggregated class |
|---|---|---|
| Intensive arable farming | Intensively farmed | Intensive agriculture |
| Intensive agro-forestry mosaic | Combinations of: Intensively farmed, intensively grass, managed forest | |
| Intensive farming | Combinations of: Intensively farmed, intensively grass | |
| Mixed farming | Combinations of: Intensively farmed, intensively grass, extensively grass | |
| Managed forestry | Managed forest | Forestry |
| Mixed forest | Combinations of: Managed forest, unmanaged forest | |
| Mixed pastoral farming | Combinations of: intensively grass, extensively grass, very extensively grass | Extensive agriculture |
| Extensive agro-forestry mosaic | Combinations of: extensively grass, very extensively grass, managed forest | |
| Peri-urban | Any combination with > 40% urban area | |
| Intensive pastoral farming | Intensively grass | Pastoral agriculture |
| Extensive pastoral farming | Extensively grass | |
| Very extensive pastoral farming | Very extensively grass | Very extensive management |
| Multifunctional | 4 or more land uses in uncommon combination | |
| Minimal management | Combinations of: very extensively grass, unmanaged forest, unmanaged land | |
| Unmanaged land | Unmanaged land | Other/no management |
| Unmanaged forest | Unmanaged forest | |
| Urban | Urban | |

**Table A1: The composition of the aggregated land use classes used here in terms of CRAFTY-EU's Agent Functional Types (AFTs) and the IAP's land use categories. In any case where the given IAP categories occupy more than 70% of a cell, that cell is allocated to the corresponding AFT in the baseline map of *CRAFTY-EU*, except in the case of the Peri-urban AFT, for which the threshold (of urban area) is 40%. The service production potentials of each AFT are calibrated to approximately match those within the IAP classes that constitute them, so that given the same productivities in a cell, the same levels of services will be produced. Names are therefore assigned in both cases on the basis of dominant land uses and do not account for minor variations in land use and production within them.**


**Appendix B: Complete territorial scenario results**


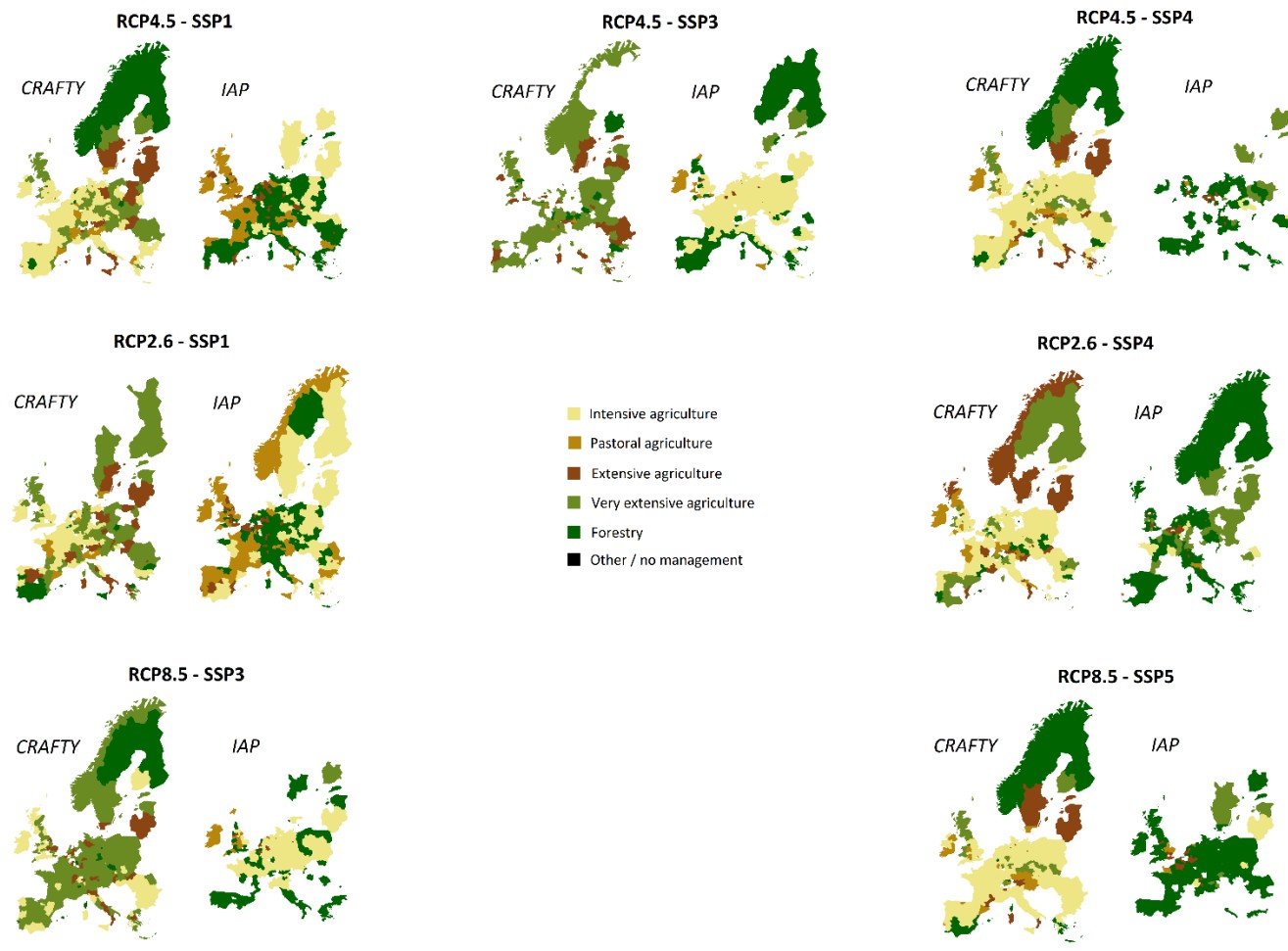

**Fig B1: Spatialised differences between the models' results for each scenario, at NUTS2 level. Colours identify the most over-represented land use type in each region in the CRAFTY and IAP results, relative to the result of the other model. White is shown where no land use type has an over-representation of more than 5% of the region's cells.**
