# Peer review of "How modelling paradigms affect simulated future land-use change"

_Earth System Dynamics, 2020_

## Referee Comment (RC1) · Helen Briassoulis (Referee) · 10 Sep 2020

**How model paradigms affect our representation of future land-use change**

**GENERAL COMMENTS**

The paper adopts the obvious and established thesis that differences in the underlying theory and assumptions of regional and larger scale LUC models (in fact, of any model) produce different projections of land use change (LUC) patterns, under alternative future scenarios, with different implications for pertinent land use decision making and planning.

It presents a comparison of two large scale pan-European integrated land use models, a constrained optimising economic-equilibrium model and a stochastic agent-based model (ABM), which the authors consider as being representative of two different modeling paradigms, under the same set of alternative future scenarios. This comparison is argued to be necessary to help model users learn about the possibilities and limitations of each modeling paradigm (and particular model) and use them intelligently.

In its present form, the main issues the paper should address are: (a) the need for a structured and articulate conceptual/theoretical framework and an associated robust methodology for model comparison to warrant the validity of the results, strengthen their analysis and support their informed, comprehensive interpretation, (b) a clear and transparent presentation (definition, explanation) and use of certain terms and (c) issues concerning the use and users of the models.

Selected important issues are detailed below that the authors might want to consider in revising the paper with the aim to improve and enhance its contribution to LUC modeling and discourse as well as to make it less esoteric and idiosyncratic and more accessible to a wider audience than it is presently the case.

WRITING STYLE. Although the writing style of the paper is acceptable, it might be written more clearly, coherently and solidly. Several comments below indicate problematic sentences and expressions.

TERMINOLOGY. The authors use, but do not explicitly define, the term 'model paradigm' to refer to the set of underlying model theory, assumptions and structure.

Given its central role in the paper, 'model paradigm' should be defined and explained.

It seems, though, that the literature uses the term 'modeling paradigm' more often. This term is encountered twice in this paper only (page 3 and page 8).

In my opinion, the term 'paradigm' should be used with caution given its strong meaning (Kuhn 1962) and should be justified based on the literature. In the present case, both models considered (optimizing and ABM), in one sense, belong to the same (reductionist) paradigm.

Based on the definition of the modeling paradigm, the paper should justify and explain why the two models compared belong to different modeling paradigms. Part of the text in Discussion and Conclusions (that concerns the models as representative of different paradigms) should be placed in the 'Methods' section.

Explain the terms: Representative Concentration Pathways (RCPs) and Shared Socioeconomic Pathways, spatial and aggregate land use change, aggregate comparison, spatial comparison.

CONCEPTUAL/THEORETICAL FRAMEWORK FOR THE COMPARISON OF MODELS. It is absolutely necessary to describe the framework/schema used for the comparison of the models; i.e. define the main factors influencing the performance and output of a model (e.g. spatial and temporal frame of reference and resolution, model specification – definition and operationalization of variables, relationships among them, missing variables –, solution techniques, data), exogenous factors and conditions, contingencies (e.g. during data collection). This conceptual framework should serve as a systematic template for the comparison (what is being compared) and constitute the basis for (a) designing the methodology and (b) presenting and interpreting the results.

COMPARISON METHODOLOGY. This should draw on the conceptual/theoretical framework and used as a basis for the design of the comparison of the models so that the factors that are not taken into account in the present comparison are held constant (controlled).

DESCRIPTION OF MODELS AND SCENARIOS. In the present form of the paper, the models and scenarios are not completely and systematically described; e.g. aim, land classes, land users, ecosystem services, other variables, etc. This makes understanding of the results and the discussion difficult.

The similarities and differences between the two models must be clearly shown (in a Table?) – variables considered, data, etc.

The limitations of ABMs should be mentioned; the emphasis on behavioural (and indirectly cultural) issues is not enough. Institutional and political issues play an important role as they are the quintessence of LUC decisions.

It should be made clear that this modeling exercise concerns food security, rural land use and associated environmental concerns. Other model aims are possible that may not be well represented by the particular models compared and/or may not yield the same projections of LUC patterns.

The instrumental assumptions (e.g. homogeneity, uniformity and similarity of land classes, land users, decision makers, etc. across Europe) of the comparison exercise should be made clear and taken into account in the discussion and interpretation of the results.

The issue of aggregation (spatial, temporal definitional) of model inputs and outputs is not discussed.

PRESENTATION OF RESULTS. At present, it is descriptive and technical and does not account for the instrumental assumptions of the comparison.

The interpretation and explanation of the results should be made within the limitations of the models (not simply noted at the end); in particular, their aggregate and reductionist nature, the very coarse spatial and temporal scale, the use of a grid that may not coincide with ecological and socio-political boundaries and conditions over the study area, the jurisdictions within decisions are made, MAUP issues, the effects of the unstructured clusters (IMPRESSIONS), instrumental assumptions, etc.

It should be underlined (and explained) that the study findings may not be applicable to lower scales.

MODEL USE AND USERS. The paper should dedicate some space (Introduction, Conclusions) to the discussion of the users and uses of these models.

Who are, or might be, the current and prospective users of these European scale aggregate models?

Are they interested in these large scales and very long time horizons, especially under conditions of serious uncertainty?

Do they have authority to make land use decisions and guide LUC at this scale and over long time horizons?

Are they really interested in providing ecosystem services? Which services in particular (use the MEA classification).

Have these models been used in real world decision making and by whom?

At some point, the authors state: "Optimising models have the advantage of representing idealised conditions". Who decides what these ideal conditions are and for whom?

**SPECIFIC COMMENTS**

TITLE

Two terms in the current title "How model paradigms affect our representation of future land-use change" may have to be modified.

Model paradigms or modeling paradigms?

Representation? The paper discusses estimates of future LUC, not representation which is something different and, in any event, it is reflected in the pre-defined land use classes and patterns used in the models.

ABSTRACT

"optimisation may be appropriate in scenarios that allow for coherent political and economic control of land systems, but not in scenarios where economic and other scenario conditions prevent the normal functioning of price signals and responses."

This is correct: model results are valid if model assumptions hold… but caution is needed here. Reality may change and render model assumptions invalid…

What is the 'normal' functioning of price signals?

"structured comparisons of parallel, transparent but paradigmatically distinct models are an important method for better understanding the potential scope and uncertainties of future land use change"

It is not clear that the present comparison is structured, or, at least, its presentation is not adequate and clear (see comments on conceptual framework and methodology).

Parallel? Explain.

Caution: understanding model results is not tantamount to understanding reality…

Who wants to understand the potential scope and uncertainties of future LUC? (the issue of users mentioned above).

"The optimisation model, in contrast, maintains food supply through intensification of agricultural production in the most profitable areas, sometimes at the expense of active management in large, contiguous parts of Europe."

This is unclear… active management? large, contiguous parts of Europe?

INTRODUCTION

"Computational models of the land system are essential in supporting efforts to limit climate change and reverse biodiversity loss (Harrison et al. 2018; Rogelj et al. 2018)"

These are not the only reasons for using these models. I suggest to broaden this sentence to encompass environmental and socio-economic change.

"The need to radically alter human land use to avert social-ecological breakdowns".

This is unclear. Is it supposed to describe the aim of planned and/or unplanned land use change? If yes, the statement should be obviously modified.

Moreover, LUC modeling is used to analyse the impacts of past and current LUC, not only to project LUC under future scenarios.

"Because other methods are not available to generate alternative findings"

Of course, there are other methods, both quantitative and qualitative (e.g. Delphi dating back to the 1970s), as well as mixed methods.

"This could be particularly misleading in social systems such as those underpinning human land use, where no universal laws or predictable patterns exist to guide model development, and modellers must instead choose between a range of contested theoretical foundations, practical designs and evaluation strategies (Brown et al. 2016; Meyfroidt et al. 45 2018; Verburg et al. 2019)."

This is a confusing sentence, especially the first part. It mixes up several notions and issues. It should be simplified and clarified.

Comment: predictable patterns are rare in nature also as complexity theory underlines and experience reveals.

"In this complex context, the proper analysis and interpretation of model outputs is just as important as proper model design"

Irrespective of context, the good modeling practice starts from the theory (however instrumental this may be) about the problem/situation modeled, that guides the development of the methodology, model design (model specification) and implementation (analysis), and, of course, the interpretation of model results.

"Steps such as standardised model descriptions, open access to model code, robust calibration and evaluation, benchmarking, uncertainty and sensitivity analyses are all necessary to ensure that model results are used appropriately (Baldos and Hertel 2013; Sohl and Claggett 2013)."

What is the meaning of the word 'used' here? Model users are not model builders and vice versa. Clarify and modify.

Moreover, the real test of a model's usefulness is model verification, not simply validation, however difficult to carry out (see, O' Sullivan).

"However, while comparisons of model outputs have been made (Lawrence et al. 2016; Prestele et al. 2016; Alexander et al. 2017), their ability to link particular outputs to particular methodological choices has been limited."

This is absolutely reasonable because the factors affecting model performance and results are interdependent, important factors may be missing and/or intangible, data are unavailable or inadequate, contingencies modify system characteristics and relationships, etc.

"Conceptual research suggests that large areas of system behaviour remain under-explored as a result (Brown et al. 2016; Huber et al. 2018; Meyfroidt et al. 2018), with the likely consequence that established findings have implicit biases and blind spots."

Yes, this is very true and should problematize efforts at comparing different models.

METHODS

This section should start with a description of the conceptual framework of the comparison.

IMPRESSIONS IAP. "Within this cross-sectoral modelling chain, rural land use is allocated within each 30-year timeslice according to a constrained optimisation algorithm that maintains equilibrium between the supply and demand for food and (as a secondary objective) timber, through iterating agricultural commodity prices (cereals, oilseeds, vegetable protein, milk, meat etc.) to promote agricultural expansion or contraction (Audsley et al. 2015)."

So, IMPRESSIONS' aim is to optimized food supply? This should be mentioned from the outset.

"similar production conditions (based on soil and agroclimate),"

Production conditions include many more factors, such as economic, fiscal, technological, institutional, etc.

"profitability thresholds used to determine which land use and management intensity is allocated to each cluster."

There are other factors (planning, policy, cultural…) that affect land use allocation and management intensity.

Moreover, caution is needed to interpret these findings given the spatio-temporal and conceptual (land use classes) aggregation of large-scale models. The 'managers' are not real people… so, what is the meaning of profitability? Profitable to whom?

"Land use proportions within each 10' x 10' grid cell represent the aggregations of the solutions for each (up to 40) associated cluster."

What is the meaning of so aggregate results?

"Modelled land manager agents compete for land on the basis of their abilities to produce a range of ecosystem services that society is assumed to require…"

How do you know that this is the aim of the land decision makers, especially at such coarse level of aggregation? I.e. to produce ecosystem services?

Psychological (emotional, political and institutional factors) regulate their relationships.

Power relations are also important determinants of land managers behaviour (however coarse their representation is).

"Land use productivities"

These should be defined taking into account the very high level of aggregation of the models.

One question is: given the high level of aggregation, how much sense do they make as goals of land managers and decision makers?

Another question is: which factors influence these very aggregate productivities?

"In CRAFTY-EU, these services are crops, meat, timber, carbon sequestration, recreation and landscape diversity. We therefore also compare ecosystem service production levels, which account for exact forms of management simulated in each cell"

The ecosystem services should be first defined in terms of the 4 main groups defined in MEA (2005) and then shown how they are operationalized.

Moreover, the sentence should be edited (crops, etc. are not services…).

"In this case, these functions are linear and equivalent for all services, meaning that the benefit of production of each service increases equally per unit of unmet demand."

A very restrictive assumption indeed…

"Importantly for this study, CRAFTY-EU is parameterised on the basis of the IAP, taking IAP outputs as exogenous conditions and replacing only the land allocation component to provide alternative land use projections under identical driving conditions."

This is unclear.

Comparison of two models when one takes input from then other?

"For ecosystem services with economic values (meat, crops and timber), agents in CRAFTY therefore make production choices consistent with this basic level of economic rationality."

First question: who are these agents at such a high level of aggregation and what is the meaning of economic rationality in this case?

Second question: what about non-economic benefits?

A note regarding NUTS2: They do not represent a uniform, EU-wide spatial division system and they differ significantly among countries.

Subsection: 3.1. Aggregate comparison

The presentation of the results is descriptive and technical.

The discussion is rather loose and tiresome with reference being made to the scenarios that have not been adequately described. The presentation of the results is rather boring and may not make a lot of sense to the reader.

One question here is: What is being interpreted? Model land use classes or real world land use?

Also, the processes that produce LUC differ between countries and for each land use class, among other factors.

"because of the gradual decision-making of agents "
What does gradual decision making mean?

"Conversely, CRAFTY responds most strongly to scenarios in which agricultural productivity decreases because its design emphasises changes in capitals that support production (climatic or socio-economic), as is particularly clear in SSP3."
The meaning is unclear.

Subsection: 3.2. Spatial comparison
This is not spatial but 'geographical' comparison because it refers to geographic areas in Europe.
The term 'spatial' is a general term and NOT identical to 'geographical'.

"In SSP4, the IAP projects substantially more very extensive and forest management than CRAFTY's more intensive results,"
extensive WHAT?
more intensive WHAT?

Subsection 3.3. Convergence experiment
The convergence/divergence of the results of different models owes to a host of factors, several of which were not examined here.
So, I wonder what is the meaning of carrying out this experiment?

4. Discussion and conclusions
I would have preferred a separate and proper Conclusions section.
The discussion of the results might be more meaningful to be combined and integrated with the presentation of results (previous section).
"Understanding the contributions of different modelling paradigms to land use projections is important for two main reasons. The first reason is that almost all large- to global-scale land system models share a single paradigm (economic optimisation of land uses). The second reason is that different paradigms are known to produce very different outcomes, but for reasons that remain unclear"
The question regarding those 'unclear reasons' is: what were the reasons hypothesized in this study? Otherwise, how was the comparison of model results carried out?

"The focused comparison presented here is therefore intended to identify and explain key differences between models representing major, distinct paradigms to project land system dynamics on the basis of complex and integrated processes founded on a small number of key, transparent assumptions".
What are these key, transparent assumptions?

Were these key differences explained? The issue of the conceptual/theoretical framework underpinning the comparison is critical here.

"An overarching distinction is apparent between the basic assumptions underlying the models. The IAP is an example of a 'topdown' model that simulates change at the system-level – in this case through an assumption of constrained economic optimisation - while CRAFTY is an example of a 'bottom-up' model that simulates change at the level of individual decisionmakers – in this case through an assumption of behavioural choices made at the level of local land systems (Brown et al. 2016).

This basic difference affects the rate, extent and pattern of simulated land use change. The consequences of top-down and bottom-up perspectives is apparent in the main forms of land use change as the models respond to scenario conditions…. "

I am not sure if any adequate explanation of the implications of different model assumptions is offered in the above excerpt.

"This difference is also apparent in our convergence experiment, where increased imports in the IAP lead to reduced agricultural area,"

But, doesn't it happen the other way around? Land is abandoned, then production drops and necessitates increase in imports… Do I miss something?

"One consequence of simulating demand and supply of a range of ecosystem services is that the relative economic support available for food production becomes a key determinant of the balance of different land uses"

What is the theoretical explanation offered?

The case may be that, because agriculture is the most extensive land use and occupies a larger number of cells, it leads to the results obtained. In other words, the results may owe to technicalities and not to real world market and social behaviour and responses.

"In both models, the simulation of the European land system as distinct from the rest of the world requires implicit assumptions about conditions in other regions and their relationships to Europe. As conceptual alternatives, therefore, neither of these necessarily capture the true dynamics of food prices and production levels, which remains a major challenge for land system modelling (Pedde et al. 2019; Müller et al. 2020)."

This is a correct remark. The exogenous factors have been incompletely modeled. Their inclusion may have further differentiated the results of the two models.

"Cell level decisions"

Do cells decide? (🙂). Which theory concerns cells?

"Indeed, their primary strength may be their ability to use theory as a guide to processes and conditions that empirical data and optimising models do not cover (Gostoli and Silverman 2020). "

This is partly true re ABMs. The question is: which theory do they use?

"The greatest value of these two approaches may therefore lie in their ability to provide alternatives; a value that is realised only in the (currently rare) cases when model assumptions are clearly communicated and when analogous models such as those used here are available for comparison (Polhill and Gotts 2009; Müller et al. 2014; Rosa et al. 2014)."

This sentence is unclear.

THE LAST PARAGRAPH of the paper is a rather unstructured list of open issues and future research directions that does not flow directly from the preceding analysis and does not offer much direction  centered around a concrete model aim…

The question is: why is it necessary to keep these modeling paradigms when alternatives are already tested and more meaningful? E.g. multi-paradigm modeling.

It might be useful to discuss in the conclusions, issues of model users, use and usefulness that might further justify pertinent future research.

**TECHNICAL COMMENTS**

Table headings should be placed at the top of the Table.

P. 23

The heading of the Table is long … it should be much shorter!

ecosystem service supply … SERVICE**S**

**SELECTED SUGGESTED READINGS**

Briassoulis, H. (2019) "Analysis of Land Use Change: Theoretical and Modeling Approaches" (2019). Web Book of Regional Science. 3. https://researchrepository.wvu.edu/rri-web-book/3

Briassoulis, H. (2001) "Policy-oriented integrated analysis of land use change: An analysis of data needs". Environmental Management, Vol. 27, No. 1: 1-11.

Briassoulis, H. (2008) «Land use policy and planning, theorizing, and modeling: Lost in translation, found in complexity?" Environment and Planning B Vol. 35: 16-33.

Couclelis, H. (2001) Modeling frameworks, paradigms and approaches. In Clarke KC, Parks BE, and Crane MP (eds) 2000. *Geographic Information Systems and Environmental Modeling*. New York: Longman & Co.

Kuhn, T. (1962, 1974) The structure of scientific revolutions.

MEA (2005) Millennium Ecosystem Assessment.

O'Sullivan D, 2004, Complexity science and human geography. *Transactions of the Institute of British Geographers*, New Series 29 282-295.

O'Sullivan D, Haklay M, 2000, ``Agent-based models and individualism: is the world agent-based? '' *Environment and Planning A* 32 1409-1425.

O'Sullivan D, Manson S, Messina J, Crawford T C, 2006, Space, place, and complexity science. *Environment and Planning A* 38: 611-617.

---

## Referee Comment (RC2) · Anonymous Referee #2 · 16 Sep 2020

Overall this is a good paper with some really interesting results. With some additional improvements to the figures/analysis it could be excellent and make an excellent contribution.

I am mostly viewing this as a scholar who uses land use projections, and the discussion of the different approaches and how they differ is really important. I like the introduction in general but a bit more detail would be appreciated.

I would also like the authors to discuss how observational data is incorperated into this models, or not. The usual standard for earth system science, is a lot of comparison to observations, so please explain how each of these paradigms try to make sure they actually compare well to observations, especially of historical land use change trajectories, or if they do not do such comparisons. If currently there is no comparison,

this could be a way to differentiate these different approaches to see which is more accurate.

I also think a bit more analysis could be helpful in the figures to synthesize a bit more. Details below.

Figure 2a: the dark (IAP) vs. lighter (CRAFTY) symbol isn't clear enough here: I recommend you make more of a matrix with left being light colors and right being dark colors and showing then that the right ones are IAP and left ones are CRAFTY. I stared at the plot for awhile before I understood what the dark grey and light grey was in the figure.

Figure 2b: the colors aren't really different enough here, and the same issue with the dark vs. light colors.

Figure 2 in general: Would a difference plot work better for this? Or a % difference? There are so many similar bar graphs?

Figure 3: white means two things here: not included, and not land? Please try use grey for one of those so this is clearer. Maybe you want difference plots here instead of these contrasting, but similar plots? Are there patterns of where in particular the differences are important that you can find and call out?

Figure 4: this graph is not self standing enough: describe what is on the left versus the right, why the arrow goes in the opposite direction on the bottom, everything needs to be explained. Describe the alternative parameterizations briefly here in the figure caption.

---

## Referee Comment (RC3) · Robert Huber (Referee) · 21 Sep 2020

I think this is a valuable contribution to the discussion on how computational models can inform political and social efforts toward more sustainable land systems on a large spatial scale. I find it very important to explicitly discuss underlying paradigms in models of land-use change and I think this contribution is an important step to improve our understanding of how the paradigm affects the interpretability and validity of such models. In my view, however, the current version of the manuscript could be improved by describing the model paradigms more explicit and by a more careful presentation of the input-output relations in the result Section. In addition, I think the contribution would gain from discussing the implications of the different paradigms to inform what the authors call "efforts to limit climate change and reverse biodiversity loss". I would

like to specify my general comment below.

1) I think it would be important to introduce the two model paradigms earlier in the manuscript. In my understanding, the first part of the discussion (lines 239-260) is something that defines the research design and should not appear as something the authors would like to discuss. In addition, I think it would help the reader if the authors would also discuss and classify/categorize these two paradigms a bit broader e.g. in the context of their own work on ABMs and their theoretical and philosophical background (Arneth et al., 2014; Brown et al., 2016).

2) In the same vein, I think that the discussion of the paradigm should also include implications from the different mathematical model implementations. If I understood the models correctly, IAP maintains equilibrium between the supply and demand for food while agents in CRAFTY compete for land-uses having a satisficing behavior including non-economic benefits. The point I'd like to make is that models based on rational economic behavior usually are characterized by switching from one corner of the mathematical solution space to another. I do not know whether this describes IAP adequately. However, the outputs seem to suggest that CRAFTY results are always more balanced than the economically driven IAP results. Thus, I would suspect that IAP jumps to corner solutions. If this is true, then this would also be known before the comparison. There might be other direct implications of the mathematical implementation of the models for the interpretation of the output. This could be introduced and discussed in the context of model paradigms.

3) I also found it difficult to follow the input-output description in the text but also the figures. The authors use the pre-defined abbreviations for the different climate and socio-economic scenarios. I understand that there are reasons not to give explicit names to these scenarios. Nevertheless, it makes the presentation of the comparison in this contribution very demanding. As a reader unfamiliar with the exact definition of each of the socio-economic scenarios, I always had to cross check what SSP3 or SSP1 now exactly implies with respect to the input assumptions. Since there is no

description of the socio-economic scenarios in Section 2.2, I had to use O'Neill et al. to be able to follow the result Section. In addition, I did not really understand how the convergence scenario was developed. I think the manuscript would profit from a concise description of the socio-economic scenarios and how these scenarios affect the underlying assumption in the model exercise e.g. production functions, demand levels etc. This would help the reader to connect input- and outputs in the different models. Personally, I would find it also helpful if there would not be abbreviations for the socio-economic scenarios. This could make it easier and more accessible to the reader e.g. in Figure 1.

4) In this context, I also had the impression that the authors did not adequately address and discuss the uncertainty with respect to model inputs. For example, the author writes that there is a "greatly increased productivity" in the scenario RCP8.5-SSP5 and consequently, the IAP model suggests that the supply of crops and meat can increase more than 30% with less than a third of the area of intensive agriculture (comparing Figure 1 and Figure 2a). The increase in productivity, in contrast, did obviously not affect land-use in CRAFTY. However, I would expect that a change in the productivity increase would considerably lower the extreme solution in the SSP5 scenario. Maybe that is something the authors wanted to address with the "convergence" comparison: Look at the sensitivity to input parameters of specific importance. I think this would deserve more attention. Maybe the authors can include more than just two input variations (increase in imports and food values) and discuss the results in the context of input uncertainty that seems to have very different impacts in the two model paradigms.

5) With respect to the methods, I acknowledge that these are well documented and state-of-art models that are suitable for comparing the effect of different model paradigms on future land-use change. However, one sentence in the manuscript confused me. The authors write (lines 113ff): "CRAFTY-EU is parameterised on the basis of the IAP, taking IAP outputs as exogenous conditions and replacing only the land allocation component to provide alternative land use projections under identical driving

conditions." What is implied here by taking the output of IAP as exogenous conditions for CRAFTY. It would not make sense to use outputs as inputs in another model and then conclude that the models have different outputs. I'm sure this is a misunderstanding (culpa mea). However, the authors should be clearer in what they do here. What are these conditions (except land-use) and how do they affect the comparison? Maybe the solution here goes hand in hand with the reply to my comment 3. However, I would suggest that the authors explain the data exchange between the models in more detail.

6)The last comment I'd like to make is probably also the most difficult to address. When looking at the results, I had the impression that the two model paradigms lead to really large differences despite using the same scenarios (e.g. in Figure 3). The authors also state that the "greatest value of these two approaches may therefore lie in their ability to provide alternatives". But if these models should inform "efforts to limit climate change and reverse biodiversity loss" what do these alternatives imply? Obviously one would come to very different conclusion what to do concerning e.g. biodiversity loss depending on the model paradigm (irrespectively of the scenario). Given the potentially contradicting (policy) conclusions from these "alternatives", critics of mathematical modelling could argue that this "invalidates" such simulations. One can get any result by choosing the "right" paradigm. I'm aware that the contribution does not attempt to address all of the caveats in model design, analysis and interpretation mentioned in the Introduction. However, I had the impression that the authors take refuge in discussing "technical integration" of models. But how could such a hybrid modelling approach solve potential contradictions? In climate change modeling, model ensembles are a way of addressing different underlying functionalities of models. However, it seems to be impossible when looking at the results of this exercise. I think this point should at least be discussed: what if model paradigms prevent instead of foster discussions on how to use modelling of more sustainable land systems on a large spatial scale? I have the impression that the authors should also discuss the value of theoretical underpinnings and conceptual frameworks (which may be more important in this context) than just "more data from another discipline on another spatial level" (which is my simplified interpretation of the

last paragraph).

Minor comments

What is the unit of the Y-axes in Figure 1? I would prefer if the difference between IAP/CRAFTY in the figures would not be represented by the level of shading only. Maybe the authors can use a different pattern or something that makes it easier to distinguish the models. I found the caption in the Figures not self-explaining (and I have to say a bit cryptic in the beginning). I do not really understand why some specific information is given in one Figure but not in the other. I think that the authors should try to make the caption self-explaining (in a way). On line 302ff, the authors state that "Conversely, (constrained) optimising models like the IAP produce idealized results that (. . .) can use flexible spatial dependencies as proxies for processes such as imitation, diffusion of knowledge or the formation of social norms (). Are you sure that knowledge diffusion and social norms fit into the economic framework of IAP? Not sure I understood this sentence.

References Arneth, A., Brown, C., Rounsevell, M.D.A., 2014. Global models of human decision-making for land-based mitigation and adaptation assessment. Nature Clim. Change 4, 550-557. Brown, C., Brown, K., Rounsevell, M., 2016. A philosophical case for process-based modelling of land use change. Modeling Earth Systems and Environment 2, 1-12.

---

## Author Comment (AC1) · 22 Oct 2020

GENERAL COMMENTS

The paper adopts the obvious and established thesis that differences in the underlying theory and assumptions of regional and larger scale LUC models (in fact, of any model) produce different projections of land use change (LUC) patterns, under alternative future scenarios, with different implications for pertinent land use decision making and planning. It presents a comparison of two large scale pan-European integrated land use models, a constrained optimising economic-equilibrium model and a stochastic agent-based model (ABM), which the authors consider as being representative of two different modeling paradigms, under the same set of alternative future scenarios.

This comparison is argued to be necessary to help model users learn about the possibilities and limitations of each modeling paradigm (and particular model) and use them intelligently. In its present form, the main issues the paper should address are: (a) the need for a structured and articulate conceptual/theoretical framework and an associated robust methodology for model comparison to warrant the validity of the results, strengthen their analysis and support their informed, comprehensive interpretation, (b) a clear and transparent presentation (definition, explanation) and use of certain terms and (c) issues concerning the use and users of the models. Selected important issues are detailed below that the authors might want to consider in revising the paper with the aim to improve and enhance its contribution to LUC modeling and discourse as well as to make it less esoteric and idiosyncratic and more accessible to a wider audience than it is presently the case.

We appreciate the detailed comments and suggestions, many thanks.

WRITING STYLE. Although the writing style of the paper is acceptable, it might be written more clearly, coherently and solidly. Several comments below indicate problematic sentences and expressions.

Thank you for identifying sections that need improvement; these suggestions will be followed in the revision and we will also carefully revise the manuscript with this comment in mind.

TERMINOLOGY. The authors use, but do not explicitly define, the term 'model paradigm' to refer to the set of underlying model theory, assumptions and structure. Given its central role in the paper, 'model paradigm' should be defined and explained. It seems, though, that the literature uses the term 'modeling paradigm' more often. This term is encountered twice in this paper only (page 3 and page 8). In my opinion, the term 'paradigm' should be used with caution given its strong meaning (Kuhn 1962) and should be justified based on the literature. In the present case, both models considered (optimizing and ABM), in one sense, belong to the same (reductionist) paradigm.

We agree that 'modelling paradigm' is more widely used and appropriate, and will adopt this term in the revision. We will also note the particular meaning of the term paradigm in the modelling literature.

Based on the definition of the modeling paradigm, the paper should justify and explain why the two models compared belong to different modeling paradigms. Part of the text in Discussion and Conclusions (that concerns the models as representative of different paradigms) should be placed in the 'Methods' section.

Good points, and we will do as suggested in the revision, introducing the models and their paradigms earlier and in a more structured way (as also suggested in the other reviews).

Explain the terms: Representative Concentration Pathways (RCPs) and Shared Socioeconomic Pathways, spatial and aggregate land use change, aggregate comparison, spatial comparison.

We will explain these terms at first usage.

CONCEPTUAL/THEORETICAL FRAMEWORK FOR THE COMPARISON OF MODELS. It is absolutely necessary to describe the framework/schema used for the comparison of the models; i.e. define the main factors influencing the performance and output of a model (e.g. spatial and temporal frame of reference and resolution, model specification – definition and operationalization of variables, relationships among them, missing variables –, solution techniques, data), exogenous factors and conditions, contingencies (e.g. during data collection). This conceptual framework should serve as a systematic template for the comparison (what is being compared) and constitute the basis for (a) designing the methodology and (b) presenting and interpreting the results.

We will describe and use this framework as suggested, adding a model comparison table to present these factors (many of which are the same or similar in both models), a model structure diagram, and additional text as necessary. We will also use the framework to structure the methods and results.

COMPARISON METHODOLOGY. This should draw on the conceptual/theoretical framework and used as a basis for the design of the comparison of the models so that the factors that are not taken into account in the present comparison are held constant (controlled).

Yes, we agree, and will present the methodology in this way in the revision.

DESCRIPTION OF MODELS AND SCENARIOS. In the present form of the paper, the models and scenarios are not completely and systematically described; e.g. aim, land classes, land users, ecosystem services, other variables, etc. This makes understanding of the results and the discussion difficult.

We are sympathetic to this point, and will include more detail in the text, new table and diagram mentioned above. We will also add new scenario descriptions. We must also note that it would be impossible to fully describe two such (previously published) models in each paper that uses them, not least because we would have to pay substantial page fees to reproduce text already available in open access publications (notably the linked article of Brown et al. 2019 in the same journal).

The similarities and differences between the two models must be clearly shown (in a Table?) – variables considered, data, etc.

We agree this would be helpful and will add a table as suggested.

The limitations of ABMs should be mentioned; the emphasis on behavioural (and indirectly cultural) issues is not enough. Institutional and political issues play an important role as they are the quintessence of LUC decisions.

We're not entirely sure what limitations are referred to here, but the absence of institutional and political behaviours is neither a universal feature of ABMs nor unique to them, and indeed they are included in this ABM but not used here. We propose to add text to the discussion to highlight their importance.

It should be made clear that this modeling exercise concerns food security, rural land use and associated environmental concerns. Other model aims are possible that may not be well represented by the particular models compared and/or may not yield the same projections of LUC patterns.

A good point, we will revise as suggested.

The instrumental assumptions (e.g. homogeneity, uniformity and similarity of land classes, land users, decision makers, etc. across Europe) of the comparison exercise should be made clear and taken into account in the discussion and interpretation of the results.

Yes, we agree this is important and will clarify and account for these assumptions (incidentally they do not include homogeneity and uniformity of land classes, as we will also clarify in the revision).

The issue of aggregation (spatial, temporal definitional) of model inputs and outputs is not discussed.

We will add explanation and discussion of these issues, in the new model overview table and the discussion section.

PRESENTATION OF RESULTS. At present, it is descriptive and technical and does not account for the instrumental assumptions of the comparison.

The interpretation and explanation of the results should be made within the limitations of the models (not simply noted at the end); in particular, their aggregate and reductionist nature, the very coarse spatial and temporal scale, the use of a grid that may not coincide with ecological and socio-political boundaries and conditions over the study area, the jurisdictions within decisions are made, MAUP issues, the effects of the unstructured clusters (IMPRESSIONS), instrumental assumptions, etc.

We will acknowledge these issues (and the other limitations) more prominently, and present and interpret the results using the framework identified above (focusing on the differences between the models, both in terms of capabilities and limitations).

It should be underlined (and explained) that the study findings may not be applicable to lower scales.

We will revise as suggested.

MODEL USE AND USERS. The paper should dedicate some space (Introduction, Conclusions) to the discussion of the users and uses of these models. Who are, or might be, the current and prospective users of these European scale aggregate models? Are they interested in these large scales and very long time horizons, especially under conditions of serious uncertainty? Do they have authority to make land use decisions and guide LUC at this scale and over long time horizons? Are they really interested in providing ecosystem services? Which services in particular (use the MEA classification). Have these models been used in real world decision making and by whom? At some point, the authors state: "Optimising models have the advantage of representing idealised conditions". Who decides what these ideal conditions are and for whom?

We will summarise these issues in the revision. The models are primarily intended and used for exploratory modelling in academic research, with the IAP in particular having also been used for capacity building (with stakeholders and students). While neither model is used directly for decision-support, there are some differences in their purposes that can inform the comparison.

SPECIFIC COMMENTS TITLE Two terms in the current title "How model paradigms affect our representation of future land-use change" may have to be modified. Model paradigms or modeling paradigms? Representation? The paper discusses estimates of future LUC, not representation which is something different and, in any event, it is reflected in the pre-defined land use classes and patterns used in the models.

Yes these are good points. We will use 'modelling paradigms' instead and change 'representation', with a new title of 'How modelling paradigms affect simulated future land-use change'

"optimisation may be appropriate in scenarios that allow for coherent political and economic control of land systems, but not in scenarios where economic and other scenario conditions prevent the

normal functioning of price signals and responses." This is correct: model results are valid if model assumptions hold… but caution is needed here. Reality may change and render model assumptions invalid…

Yes we entirely agree and will reword to clarify our meaning.

What is the 'normal' functioning of price signals?

This was poorly phrased. We will use 'equilibrium' instead.

"structured comparisons of parallel, transparent but paradigmatically distinct models are an important method for better understanding the potential scope and uncertainties of future land use change" It is not clear that the present comparison is structured, or, at least, its presentation is not adequate and clear (see comments on conceptual framework and methodology). Parallel? Explain. Caution: understanding model results is not tantamount to understanding reality… Who wants to understand the potential scope and uncertainties of future LUC? (the issue of users mentioned above).

The structure of the comparison will be clarified in the revision as described above, also explaining in detail the respects in which the models are 'parallel'. We agree that model results are distinct from reality, and will deal with their interpretation in the new text on model uses/users.

"The optimisation model, in contrast, maintains food supply through intensification of agricultural production in the most profitable areas, sometimes at the expense of active management in large, contiguous parts of Europe."

This is unclear… active management? large, contiguous parts of Europe?

We will rephrase for clarity.

INTRODUCTION
"Computational models of the land system are essential in supporting efforts to limit climate change and reverse biodiversity loss (Harrison et al. 2018; Rogelj et al. 2018)"
These are not the only reasons for using these models. I suggest to broaden this sentence to encompass environmental and socio-economic change.

We will revise as suggested.

"The need to radically alter human land use to avert social-ecological breakdowns".
This is unclear. Is it supposed to describe the aim of planned and/or unplanned land use change? If yes, the statement should be obviously modified.
Moreover, LUC modeling is used to analyse the impacts of past and current LUC, not only to project LUC under future scenarios.

This sentence refers simply to the unsustainable impacts of current land use. We will clarify this and mention other model uses in an additional sentence in the revision.

"Because other methods are not available to generate alternative findings"
Of course, there are other methods, both quantitative and qualitative (e.g. Delphi dating back to the 1970s), as well as mixed methods.

Yes this was poorly phrased; we will clarify that modelling provides one approach to exploring possible future changes that, while useful, is not the only approach possible.

"This could be particularly misleading in social systems such as those underpinning human land use, where no universal laws or predictable patterns exist to guide model development, and modellers

must instead choose between a range of contested theoretical foundations, practical designs and evaluation strategies (Brown et al. 2016; Meyfroidt et al. 45 2018; Verburg et al. 2019)."
This is a confusing sentence, especially the first part. It mixes up several notions and issues. It should be simplified and clarified.
Comment: predictable patterns are rare in nature also as complexity theory underlines and experience reveals.

We will split and revise this sentence. We agree that predictability is rare in natural systems, but regard it as a particularly important issue in human/social systems where human behaviour introduces extra challenges beyond complexity.

"In this complex context, the proper analysis and interpretation of model outputs is just as important as proper model design"
Irrespective of context, the good modeling practice starts from the theory (however instrumental this may be) about the problem/situation modeled, that guides the development of the methodology, model design (model specification) and implementation (analysis), and, of course, the interpretation of model results.

We agree, and will emphasise that the current study aims to improve interpretation.

"Steps such as standardised model descriptions, open access to model code, robust calibration and evaluation, benchmarking, uncertainty and sensitivity analyses are all necessary to ensure that model results are used appropriately (Baldos and Hertel 2013; Sohl and Claggett 2013)."
What is the meaning of the word 'used' here? Model users are not model builders and vice versa. Clarify and modify. Moreover, the real test of a model's usefulness is model verification, not simply validation, however difficult to carry out (see, O' Sullivan).

We will change 'used' to 'interpreted'. Model uses certainly extend beyond the uses model builders put them to, and accurate interpretation underpins them all. We also agree that model verification is crucial and involves more than the steps we mention here, and will amend the sentence to reflect this.

"However, while comparisons of model outputs have been made (Lawrence et al. 2016; Prestele et al. 2016; Alexander et al. 2017), their ability to link particular outputs to particular methodological choices has been limited."
This is absolutely reasonable because the factors affecting model performance and results are interdependent, important factors may be missing and/or intangible, data are unavailable or inadequate, contingencies modify system characteristics and relationships, etc.

There are certainly limits to linking model design to model outputs, but we would suggest that these limits have not yet been reached, partly because model comparisons have been relatively few and relatively limited in their scope, for instance not including detailed comparisons of quite distinct but parallel models (in the sense of sharing application coverage, resolution, contextual data etc.) of the kind that we make here. We will more clearly present this as the motivation for the study in the revision, and carefully define the basis of the comparison in terms of model similarities and differences.

"Conceptual research suggests that large areas of system behaviour remain under-explored as a result (Brown et al. 2016; Huber et al. 2018; Meyfroidt et al. 2018), with the likely consequence that established findings have implicit biases and blind spots."
Yes, this is very true and should problematize efforts at comparing different models.

We will extend this point to acknowledge that comparisons can illuminate some but not all such biases and blind spots.

METHODS
This section should start with a description of the conceptual framework of the comparison.

We will revise as suggested.

IMPRESSIONS IAP. "Within this cross-sectoral modelling chain, rural land use is allocated within each 30-year timeslice according to a constrained optimisation algorithm that maintains equilibrium between the supply and demand for food and (as a secondary objective) timber, through iterating agricultural commodity prices (cereals, oilseeds, vegetable protein, milk, meat etc.) to promote agricultural expansion or contraction (Audsley et al. 2015)."
So, IMPRESSIONS' aim is to optimized food supply? This should be mentioned from the outset.

We will clarify from the outset that the model aims to satisfy food demand (taking account of net imports), and does so optimally subject to constraints imposed by biophysical and socio-economic conditions.

"similar production conditions (based on soil and agroclimate),"
Production conditions include many more factors, such as economic, fiscal, technological, institutional, etc.

Yes, we were referring more specifically to biophysical conditions here, and will amend as such ('similar biophysical conditions (based on soil and agroclimate)').

"profitability thresholds used to determine which land use and management intensity is allocated to each cluster."
There are other factors (planning, policy, cultural…) that affect land use allocation and management intensity.
Moreover, caution is needed to interpret these findings given the spatio-temporal and conceptual (land use classes) aggregation of large-scale models. The 'managers' are not real people… so, what is the meaning of profitability? Profitable to whom?

We will describe land allocation in both models more fully in the revision, and in particular the role of these and other additional factors (many of which affect allocation). Profitability is used here to mean the simulated profit available for a particular level of production in a particular cell, and indeed does not refer to profit to real land managers, which we will emphasise.

"Land use proportions within each 10' x 10' grid cell represent the aggregations of the solutions for each (up to 40) associated cluster."
What is the meaning of so aggregate results?

We will clarify the derivation and interpretation of the aggregated results. The aggregation is the result of spatial weighting of the optimised land use solution for each cluster containing the grid cell in question. The clustering recognises that different biophysical conditions (soil and agroclimate) differentially influence the suitability, productivity and profitability of different crops and different agricultural systems (arable, dairy etc.), leading to heterogeneity in agricultural land use within a grid cell.

"Modelled land manager agents compete for land on the basis of their abilities to produce a range of ecosystem services that society is assumed to require…"
How do you know that this is the aim of the land decision makers, especially at such coarse level of aggregation? I.e. to produce ecosystem services?
Psychological (emotional, political and institutional factors) regulate their relationships.
Power relations are also important determinants of land managers behaviour (however coarse their representation is).

This text describes processes in the model, as distinct from reality – the competition for land on the basis of ecosystem service provision is a modelling assumption analogous to the allocation of land uses on the basis of profitability in the IAP. We will describe the inclusion/exclusion of particular factors in more detail in the new model description table and also address missing factors and processes in the text.

"Land use productivities"
These should be defined taking into account the very high level of aggregation of the models.
One question is: given the high level of aggregation, how much sense do they make as goals of land managers and decision makers?
Another question is: which factors influence these very aggregate productivities?

We will rephrase for clarity (this refers to the yields / ecosystem service provision levels of the different simulated land use systems (crops, grassland, forestry etc.) under the agronomic scenario conditions.

"In CRAFTY-EU, these services are crops, meat, timber, carbon sequestration, recreation and landscape diversity. We therefore also compare ecosystem service production levels, which account for exact forms of management simulated in each cell"
The ecosystem services should be first defined in terms of the 4 main groups defined in MEA (2005) and then shown how they are operationalized.
Moreover, the sentence should be edited (crops, etc. are not services…).

We will revise as suggested.

"In this case, these functions are linear and equivalent for all services, meaning that the benefit of production of each service increases equally per unit of unmet demand."
A very restrictive assumption indeed…

This is deliberately restrictive, yes. It allows us to compare an equal weighting of service provision (in CRAFTY) with a focus on food production (in the IAP), and avoids a more complex but equally arbitrary weighting that would make results harder to interpret.

"Importantly for this study, CRAFTY-EU is parameterised on the basis of the IAP, taking IAP outputs as exogenous conditions and replacing only the land allocation component to provide alternative land use projections under identical driving conditions."
This is unclear.
Comparison of two models when one takes input from then other?

We will clarify, and describe the inputs and outputs of each model, and their relationships to one another, in the new table and diagram described above.

"For ecosystem services with economic values (meat, crops and timber), agents in CRAFTY therefore make production choices consistent with this basic level of economic rationality."
First question: who are these agents at such a high level of aggregation and what is the meaning of economic rationality in this case?
Second question: what about non-economic benefits?

We will revise this sentence for clarity. As described in the text, modelled agents do not correspond to real-world actors, but are used to capture elements of their behaviour within localised land systems. Non-economic benefits are also included in CRAFTY.

A note regarding NUTS2: They do not represent a uniform, EU-wide spatial division system and they differ significantly among countries.

We agree. There is no ideal resolution at which to make this comparison, but we chose NUTS2 as an established system to complement the results we present at cell and European scales.

Subsection: 3.1. Aggregate comparison
The presentation of the results is descriptive and technical.
The discussion is rather loose and tiresome with reference being made to the scenarios that have not been adequately described. The presentation of the results is rather boring and may not make a lot of sense to the reader.

We will add a description of the scenarios in the revision. We will also edit the text to ensure clarity and interest where we can.

One question here is: What is being interpreted? Model land use classes or real world land use? Also, the processes that produce LUC differ between countries and for each land use class, among other factors.

Model land use classes are being interpreted. We will clarify this prominently in describing the conceptual framework of the exercise. We agree that processes differ and will add discussion of this important point.

"because of the gradual decision-making of agents "
What does gradual decision making mean?

This refers to agents' decisions having some probability of being delayed across multiple timesteps (representing years), rather than taking immediate effect. We will explain this in the text.

"Conversely, CRAFTY responds most strongly to scenarios in which agricultural productivity decreases because its design emphasises changes in capitals that support production (climatic or socio-economic), as is particularly clear in SSP3."
The meaning is unclear.

To be rephrased.

Subsection: 3.2. Spatial comparison
This is not spatial but 'geographical' comparison because it refers to geographic areas in Europe.
The term 'spatial' is a general term and NOT identical to 'geographical'.

We will label this as a geographical comparison.

"In SSP4, the IAP projects substantially more very extensive and forest management than CRAFTY's more intensive results,"
extensive WHAT?
more intensive WHAT?

To be revised as 'extensive agricultural management' and 'intensive agricultural management'.

Subsection 3.3. Convergence experiment
The convergence/divergence of the results of different models owes to a host of factors, several of which were not examined here.
So, I wonder what is the meaning of carrying out this experiment?

We will clarify the purpose of this experiment in the text: it is indeed intended to identify these factors in this particular case. The observed divergence in this scenario is partly due to conditions differing in the models (because food prices rise higher in the IAP and production levels fall lower in CRAFTY). By controlling these differences, we are able to identify additional factors that cause

divergence – and in this case they reflect basic modelling assumptions, the effects of which would otherwise remain obscure.

4. Discussion and conclusions
I would have preferred a separate and proper Conclusions section.
The discussion of the results might be more meaningful to be combined and integrated with the presentation of results (previous section).

We will produce a stand-alone Conclusions section but prefer to keep results and discussion separate as we find it important to establish technical findings before interpreting them.

"Understanding the contributions of different modelling paradigms to land use projections is important for two main reasons. The first reason is that almost all large- to global-scale land system models share a single paradigm (economic optimisation of land uses). The second reason is that different paradigms are known to produce very different outcomes, but for reasons that remain unclear"
The question regarding those 'unclear reasons' is: what were the reasons hypothesized in this study? Otherwise, how was the comparison of model results carried out?

In this case, we hypothesise that the decision making / allocation paradigm is one dominant source of uncertainty in land use modelling, as opposed to uncertainty in crop yields, biophysical conditions etc. Hence we keep the latter factors common between the models and explore how different factors that influence decision making (profitability; demand; socio-economic conditions) affect the models. We will set this hypothesis out in the revision as described above and use it to structure the methods and results section.

"The focused comparison presented here is therefore intended to identify and explain key differences between models representing major, distinct paradigms to project land system dynamics on the basis of complex and integrated processes founded on a small number of key, transparent assumptions".
What are these key, transparent assumptions? Were these key differences explained? The issue of the conceptual/theoretical framework underpinning the comparison is critical here.

We will elucidate the underlying framework in the revision with particular emphasis on these key assumptions that differ between the models.

"An overarching distinction is apparent between the basic assumptions underlying the models. The IAP is an example of a 'topdown' model that simulates change at the system-level – in this case through an assumption of constrained economic optimisation - while CRAFTY is an example of a 'bottom-up' model that simulates change at the level of individual decisionmakers – in this case through an assumption of behavioural choices made at the level of local land systems (Brown et al. 2016).
This basic difference affects the rate, extent and pattern of simulated land use change. The consequences of top-down and bottom-up perspectives is apparent in the main forms of land use change as the models respond to scenario conditions…. "
I am not sure if any adequate explanation of the implications of different model assumptions is offered in the above excerpt.

Explanation of implications is in the text that follows the quoted sentences. We believe the additional structured comparison of the models in the revision will help to explain the assumptions and their implications.

"This difference is also apparent in our convergence experiment, where increased imports in the IAP lead to reduced agricultural area,"

But, doesn't it happen the other way around? Land is abandoned, then production drops and necessitates increase in imports… Do I miss something?

We will clarify this. The convergence experiment involved pre-emptively increasing imports in the IAP to mimic the lower European production levels generated by CRAFTY.

"One consequence of simulating demand and supply of a range of ecosystem services is that the relative economic support available for food production becomes a key determinant of the balance of different land uses"
What is the theoretical explanation offered?
The case may be that, because agriculture is the most extensive land use and occupies a larger number of cells, it leads to the results obtained. In other words, the results may owe to technicalities and not to real world market and social behaviour and responses.

An interesting point, and one we will address in the revision. The result is certainly due to technicalities, in the sense that the model is sensitive to the relative valuation because that is the basis for simulated land competition, but reflects the reality that land use, as primarily economically-driven, is subject to the relative economic support for food production and for other ecosystem service provision.

"In both models, the simulation of the European land system as distinct from the rest of the world requires implicit assumptions about conditions in other regions and their relationships to Europe. As conceptual alternatives, therefore, neither of these necessarily capture the true dynamics of food prices and production levels, which remains a major challenge for land system modelling (Pedde et al. 2019; Müller et al. 2020)."
This is a correct remark. The exogenous factors have been incompletely modeled. Their inclusion may have further differentiated the results of the two models.

Yes, we agree. We will add a sentence to emphasise that different approaches to modelling exogenous factors would likely introduce even greater differences.

"Cell level decisions"
Do cells decide? ( ). Which theory concerns cells?

We will rephrase as 'simulated decisions affecting individual cells'

"Indeed, their primary strength may be their ability to use theory as a guide to processes and conditions that empirical data and optimising models do not cover (Gostoli and Silverman 2020). "
This is partly true re ABMs. The question is: which theory do they use?

We will add a comment emphasising the importance of the choice of theory.

"The greatest value of these two approaches may therefore lie in their ability to provide alternatives; a value that is realised only in the (currently rare) cases when model assumptions are clearly communicated and when analogous models such as those used here are available for comparison (Polhill and Gotts 2009; Müller et al. 2014; Rosa et al. 2014)."
This sentence is unclear.

We will rephrase this sentence.

THE LAST PARAGRAPH of the paper is a rather unstructured list of open issues and future research directions that does not flow directly from the preceding analysis and does not offer much direction centered around a concrete model aim…

To be replaced with a Conclusion section.

The question is: why is it necessary to keep these modeling paradigms when alternatives are already tested and more meaningful? E.g. multi-paradigm modeling.

We are not entirely sure in what sense 'multi-paradigm modeling' is being used here, but the models and paradigms represented here have also been tested and found to be meaningful in a number of ways. In any case, to the extent that different paradigms are present within 'multi-paradigm modeling', our basic premise of understanding how underlying assumptions influence model outputs is still relevant.

It might be useful to discuss in the conclusions, issues of model users, use and usefulness that might further justify pertinent future research

Yes, we find this a good suggestion and will add some discussion.

**TECHNICAL COMMENTS**
Table headings should be placed at the top of the Table.
P. 23
The heading of the Table is long … it should be much shorter!
ecosystem service supply … SERVICES

Changes to be made.

---

## Author Comment (AC2) · 22 Oct 2020

**Reviewer 2:**

Overall this is a good paper with some really interesting results. With some additional improvements to the figures/analysis it could be excellent and make an excellent contribution.

Thanks for the positive comments and the suggestions.

I am mostly viewing this as a scholar who uses land use projections, and the discussion of the different approaches and how they differ is really important. I like the introduction in general but a bit more detail would be appreciated.

We will add more detail, in common with the responses to R1.

I would also like the authors to discuss how observational data is incorperated into this models, or not. The usual standard for earth system science, is a lot of comparison to observations, so please explain how each of these paradigms try to make sure they actually compare well to observations, especially of historical land use change trajectories, or if they do not do such comparisons. If currently there is no comparison, this could be a way to differentiate these different approaches to see which is more accurate.

Thanks, an important point. We will detail specific and general use of observational data in the revision, particularly in the model description/comparison table referred to in responses to Reviewer 1. In general, optimising models have indeed been more often compared against data but we will provide specific details for these models.

I also think a bit more analysis could be helpful in the figures to synthesize a bit more. Details below.
Figure 2a: the dark (IAP) vs. lighter (CRAFTY) symbol isn't clear enough here: I recommend you make more of a matrix with left being light colors and right being dark colors and showing then that the right ones are IAP and left ones are CRAFTY. I stared at the plot for awhile before I understood what the dark grey and light grey was in the figure.

Figure 2b: the colors aren't really different enough here, and the same issue with the dark vs. light colors.
Figure 2 in general: Would a difference plot work better for this? Or a % difference? There are so many similar bar graphs?
Figure 3: white means two things here: not included, and not land? Please try use grey for one of those so this is clearer. Maybe you want difference plots here instead of these contrasting, but similar plots? Are there patterns of where in particular the differences are important that you can find and call out?
Figure 4: this graph is not self standing enough: describe what is on the left versus the right, why the arrow goes in the opposite direction on the bottom, everything needs to be explained. Describe the alternative parameterizations briefly here in the figure caption.

We will make all of these changes as suggested.

---

## Author Comment (AC3) · 22 Oct 2020

**Reviewer 3 (Robert Huber):**

I think this is a valuable contribution to the discussion on how computational models can inform political and social efforts toward more sustainable land systems on a large spatial scale. I find it very important to explicitly discuss underlying paradigms in models of land-use change and I think this contribution is an important step to improve our understanding of how the paradigm affects the interpretability and validity of such models. In my view, however, the current version of the manuscript could be improved by describing the model paradigms more explicit and by a more careful presentation of the input-output relations in the result Section. In addition, I think the contribution would gain from discussing the implications of the different paradigms to inform what the authors call "efforts to limit climate change and reverse biodiversity loss". I would like to specify my general comment below.

Many thanks for the positive feedbacks and useful suggestions, all of which we propose to adopt.

1) I think it would be important to introduce the two model paradigms earlier in the manuscript. In my understanding, the first part of the discussion (lines 239-260) is something that defines the research design and should not appear as something the authors would like to discuss. In addition, I think it would help the reader if the authors would also discuss and classify/categorize these two paradigms a bit broader e.g. in the context of their own work on ABMs and their theoretical and philosophical background (Arneth et al., 2014; Brown et al., 2016).

We will follow this suggestion, moving the text referred to back in the manuscript, detailing the paradigms more carefully in the new model descriptions and relating them to earlier publications as suggested.

2) In the same vein, I think that the discussion of the paradigm should also include implications from the different mathematical model implementations. If I understood the models correctly, IAP maintains equilibrium between the supply and demand for food while agents in CRAFTY compete for land-uses having a satisficing behavior including non-economic benefits. The point I'd like to make is that models based on rational economic behavior usually are characterized by switching from one corner of the mathematical solution space to another. I do not know whether this describes IAP adequately. However, the outputs seem to suggest that CRAFTY results are always more balanced than the economically driven IAP results. Thus, I would suspect that IAP jumps to corner solutions. If this is true, then this would also be known before the comparison. There might be other direct implications of the mathematical implementation of the models for the interpretation of the output. This could be introduced and discussed in the context of model paradigms.

A good point, and we will highlight the issues of this sort that are clear prior to the comparison, with those relating to these specific models in the model comparison table and those relating to these paradigms more generally in new text. This attribution is slightly complex – for instance the IAP simulates and aggregates up to 40 clusters in each grid cell that produce different solutions, with changes also possible within a land use class (e.g. different crop selections), but there are certainly elements of model design that mathematically pre-define model outcomes, yes.

3) I also found it difficult to follow the input-output description in the text but also the figures. The authors use the pre-defined abbreviations for the different climate and socio-economic scenarios. I understand that there are reasons not to give explicit names to these scenarios. Nevertheless, it makes the presentation of the comparison in this contribution very demanding. As a reader unfamiliar with the exact definition of each of the socio-economic scenarios, I always had to cross check what SSP3 or SSP1 now exactly implies with respect to the input assumptions. Since there is no description of the socio-economic scenarios in Section 2.2, I had to use O'Neill et al. to be able to

follow the result Section. In addition, I did not really understand how the convergence scenario was developed. I think the manuscript would profit from a concise description of the socio-economic scenarios and how these scenarios affect the underlying assumption in the model exercise e.g. production functions, demand levels etc. This would help the reader to connect input- and outputs in the different models. Personally, I would find it also helpful if there would not be abbreviations for the socio-economic scenarios. This could make it easier and more accessible to the reader e.g. in Figure 1.

Thanks for highlighting this. We will add a scenario description and implementation table, and also avoid the acronyms where possible (while perhaps keeping them where the full names would otherwise be frequently repeated).

4) In this context, I also had the impression that the authors did not adequately address and discuss the uncertainty with respect to model inputs. For example, the author writes that there is a "greatly increased productivity" in the scenario RCP8.5-SSP5 and consequently, the IAP model suggests that the supply of crops and meat can increase more than 30% with less than a third of the area of intensive agriculture (comparing Figure 1 and Figure 2a). The increase in productivity, in contrast, did obviously not affect land-use in CRAFTY. However, I would expect that a change in the productivity increase would considerably lower the extreme solution in the SSP5 scenario. Maybe that is something the authors wanted to address with the "convergence" comparison: Look at the sensitivity to input parameters of specific importance. I think this would deserve more attention. Maybe the authors can include more than just two input variations (increase in imports and food values) and discuss the results in the context of input uncertainty that seems to have very different impacts in the two model paradigms.

An interesting point, and it's quite correct that we haven't dealt with uncertainty/sensitivity in any depth here – we agree that we should include more on this.  While we're wary of adding more experiments here in addition to substantial extra explanation as suggested by the reviewers, both models have been quite extensively assessed in sensitivity and uncertainty analyses in the past, including with respect to scenario conditions, and we therefore propose to include a summary of these findings and particularly their bearing on the differences between the models that we identify. We believe this would indeed improve interpretation of the findings.

5) With respect to the methods, I acknowledge that these are well documented and state-of-art models that are suitable for comparing the effect of different model paradigms on future land-use change. However, one sentence in the manuscript confused me. The authors write (lines 113ff): "CRAFTY-EU is parameterised on the basis of the IAP, taking IAP outputs as exogenous conditions and replacing only the land allocation component to provide alternative land use projections under identical driving conditions." What is implied here by taking the output of IAP as exogenous conditions for CRAFTY. It would not make sense to use outputs as inputs in another model and then conclude that the models have different outputs. I'm sure this is a misunderstanding (culpa mea). However, the authors should be clearer in what they do here. What are these conditions (except land-use) and how do they affect the comparison? Maybe the solution here goes hand in hand with the reply to my comment 3. However, I would suggest that the authors explain the data exchange between the models in more detail.

Yes agreed, we will explain this properly in the revision, including via a model diagram that shows the relationship of the two models and their input/output sharing.

6)The last comment I'd like to make is probably also the most difficult to address. When looking at the results, I had the impression that the two model paradigms lead to really large differences despite using the same scenarios (e.g. in Figure 3). The authors also state that the "greatest value of

these two approaches may therefore lie in their ability to provide alternatives". But if these models should inform "efforts to limit climate change and reverse biodiversity loss" what do these alternatives imply? Obviously one would come to very different conclusion what to do concerning e.g. biodiversity loss depending on the model paradigm (irrespectively of the scenario). Given the potentially contradicting (policy) conclusions from these "alternatives", critics of mathematical modelling could argue that this "invalidates" such simulations. One can get any result by choosing the "right" paradigm. I'm aware that the contribution does not attempt to address all of the caveats in model design, analysis and interpretation mentioned in the Introduction. However, I had the impression that the authors take refuge in discussing "technical integration" of models. But how could such a hybrid modelling approach solve potential contradictions? In climate change modeling, model ensembles are a way of addressing different underlying functionalities of models. However, it seems to be impossible when looking at the results of this exercise. I think this point should at least be discussed: what if model paradigms prevent instead of foster discussions on how to use modelling of more sustainable land systems on a large spatial scale? I have the impression that the authors should also discuss the value of theoretical underpinnings and conceptual frameworks (which may be more important in this context) than just "more data from another discipline on another spatial level" (which is my simplified interpretation of the last paragraph).

We find these excellent suggestions and fair criticisms. It is probably true that we take refuge in technical issues to some extent! This is partly because we wish to establish basic differences here, but we should have better addressed this overarching issue. We will therefore add text in the discussion to link our findings to the motivating question of model uses, and actually believe we can suggest some useful ways forward in terms of converging on more balanced representations that account for the different effects highlighted in the comparison.

Minor comments

What is the unit of the Y-axes in Figure 1? I would prefer if the difference between IAP/CRAFTY in the figures would not be represented by the level of shading only. Maybe the authors can use a different pattern or something that makes it easier to distinguish the models. I found the caption in the Figures not self-explaining (and I have to say a bit cryptic in the beginning). I do not really understand why some specific information is given in one Figure but not in the other. I think that the authors should try to make the caption self-explaining (in a way). On line 302ff, the authors state that "Conversely, (constrained) optimising models like the IAP produce idealized results that (. . .) can use flexible spatial dependencies as proxies for processes such as imitation, diffusion of knowledge or the formation of social norms (). Are you sure that knowledge diffusion and social norms fit into the economic framework of IAP? Not sure I understood this sentence.

All to be changed as suggested.

---

## Author Response (AR1)

**Please note that line numbers refer to the 'track changes' version of the manuscript.**

**Reviewer 1 – Helen Briassoulis**

**GENERAL COMMENTS**

The paper adopts the obvious and established thesis that differences in the underlying theory and assumptions of regional and larger scale LUC models (in fact, of any model) produce different projections of land use change (LUC) patterns, under alternative future scenarios, with different implications for pertinent land use decision making and planning. It presents a comparison of two large scale pan-European integrated land use models, a constrained optimising economic-equilibrium model and a stochastic agent-based model (ABM), which the authors consider as being representative of two different modeling paradigms, under the same set of alternative future scenarios.

This comparison is argued to be necessary to help model users learn about the possibilities and limitations of each modeling paradigm (and particular model) and use them intelligently. In its present form, the main issues the paper should address are: (a) the need for a structured and articulate conceptual/theoretical framework and an associated robust methodology for model comparison to warrant the validity of the results, strengthen their analysis and support their informed, comprehensive interpretation, (b) a clear and transparent presentation (definition, explanation) and use of certain terms and (c) issues concerning the use and users of the models. Selected important issues are detailed below that the authors might want to consider in revising the paper with the aim to improve and enhance its contribution to LUC modeling and discourse as well as to make it less esoteric and idiosyncratic and more accessible to a wider audience than it is presently the case.

**We appreciate the detailed comments and suggestions, many thanks.**

WRITING STYLE. Although the writing style of the paper is acceptable, it might be written more clearly, coherently and solidly. Several comments below indicate problematic sentences and expressions.

**Thank you for identifying sections that need improvement; these suggestions have been followed in the revision and we also carefully revised the manuscript with this comment in mind.**

TERMINOLOGY. The authors use, but do not explicitly define, the term 'model paradigm' to refer to the set of underlying model theory, assumptions and structure. Given its central role in the paper, 'model paradigm' should be defined and explained. It seems, though, that the literature uses the term 'modeling paradigm' more often. This term is encountered twice in this paper only (page 3 and page 8). In my opinion, the term 'paradigm' should be used with caution given its strong meaning (Kuhn 1962) and should be justified based on the literature. In the present case, both models considered (optimizing and ABM), in one sense, belong to the same (reductionist) paradigm.

**We agree that 'modelling paradigm' is more widely used and appropriate, and have adopted this term throughout the revision. We have also noted the particular meaning of the term paradigm in the modelling literature (lines 75-77; 192-200).**

Based on the definition of the modeling paradigm, the paper should justify and explain why the two models compared belong to different modeling paradigms. Part of the text in Discussion and Conclusions (that concerns the models as representative of different paradigms) should be placed in the 'Methods' section.

Good points, and have done as suggested in the revision, introducing the models and their paradigms earlier and in a more structured way (as also suggested in the other reviews) (in particular lines 185-200, also 75-77).

Explain the terms: Representative Concentration Pathways (RCPs) and Shared Socioeconomic Pathways, spatial and aggregate land use change, aggregate comparison, spatial comparison.

We have explained or altered each of these terms.

CONCEPTUAL/THEORETICAL FRAMEWORK FOR THE COMPARISON OF MODELS. It is absolutely necessary to describe the framework/schema used for the comparison of the models; i.e. define the main factors influencing the performance and output of a model (e.g. spatial and temporal frame of reference and resolution, model specification – definition and operationalization of variables, relationships among them, missing variables –, solution techniques, data), exogenous factors and conditions, contingencies (e.g. during data collection). This conceptual framework should serve as a systematic template for the comparison (what is being compared) and constitute the basis for (a) designing the methodology and (b) presenting and interpreting the results.

We have described and used this framework as suggested, adding a model comparison table (the new Table 3) to present these factors (many of which are the same or similar in both models), a model structure diagram (the new Figure 1), and additional text as necessary (in particular Tables 1 and 2 in addition to Table 3, and lines 372-379, 418-425 amongst further clarifications detailed below). We have also used the framework to structure the methods and results (lines 185-218, 239-244).

COMPARISON METHODOLOGY. This should draw on the conceptual/theoretical framework and used as a basis for the design of the comparison of the models so that the factors that are not taken into account in the present comparison are held constant (controlled).

**Yes, we agree, and have presented the methodology in this way in the revision (key text in lines 186-191, 239-244).**

DESCRIPTION OF MODELS AND SCENARIOS. In the present form of the paper, the models and scenarios are not completely and systematically described; e.g. aim, land classes, land users, ecosystem services, other variables, etc. This makes understanding of the results and the discussion difficult.

We are sympathetic to this point, and have included substantially more detail in the text, new tables (1-3) and diagram (Fig. 1) mentioned above. We have also added new scenario descriptions in the new Table 1. We must also note that it would be impossible to fully describe two such (previously published) models in each paper that uses them, not least because we would have to pay substantial page fees to reproduce text already available in open access publications (notably the linked article of Brown et al. 2019 in the same journal).

The similarities and differences between the two models must be clearly shown (in a Table?) – variables considered, data, etc.

We agree this would be helpful and have added a table as suggested (Table 3, with key differences in outcomes in Table 1).

The limitations of ABMs should be mentioned; the emphasis on behavioural (and indirectly cultural) issues is not enough. Institutional and political issues play an important role as they are the quintessence of LUC decisions.

We're not entirely sure what limitations are referred to here, but the absence of institutional and political behaviours is neither a universal feature of ABMs nor unique to them, and indeed they are included in this ABM but not used here (clarified in lines 133-136). We have added text to highlight the importance of 'missing' factors (lines 428-430, 439-442).

It should be made clear that this modeling exercise concerns food security, rural land use and associated environmental concerns. Other model aims are possible that may not be well represented by the particular models compared and/or may not yield the same projections of LUC patterns.

**A good point, and we have revised as suggested (lines 16-17 and added relevant details throughout as described above).**

The instrumental assumptions (e.g. homogeneity, uniformity and similarity of land classes, land users, decision makers, etc. across Europe) of the comparison exercise should be made clear and taken into account in the discussion and interpretation of the results.

Yes, we agree this is important and have clarified and accounted for these assumptions (incidentally they do not include homogeneity and uniformity of land classes, as also clarified) (Table 3, lines 123-127, 223-225, 345-346, 428-430, 439-442).

The issue of aggregation (spatial, temporal definitional) of model inputs and outputs is not discussed.

We have added explanation and discussion of these issues in Table 3 and lines 223-225, in addition to Tables 4 and A1.

PRESENTATION OF RESULTS. At present, it is descriptive and technical and does not account for the instrumental assumptions of the comparison.

The interpretation and explanation of the results should be made within the limitations of the models (not simply noted at the end); in particular, their aggregate and reductionist nature, the very coarse spatial and temporal scale, the use of a grid that may not coincide with ecological and socio-political boundaries and conditions over the study area, the jurisdictions within decisions are made, MAUP issues, the effects of the unstructured clusters (IMPRESSIONS), instrumental assumptions, etc.

We have acknowledged and provided more detail about these issues (and other limitations) more prominently (Table 3, new text throughout the Methods and Discussion sections, and in particular in the new section 2.3), and present and interpret the results using the framework identified above (focusing on the differences between the models, both in terms of capabilities and limitations).

It should be underlined (and explained) that the study findings may not be applicable to lower scales.

Revised as suggested (lines 428-430, 439-442, as well as clarification of resolution issues in Table 3 and elsewhere in the text).

MODEL USE AND USERS. The paper should dedicate some space (Introduction, Conclusions) to the discussion of the users and uses of these models. Who are, or might be, the current and prospective users of these European scale aggregate models? Are they interested in these large scales and very long time horizons, especially under conditions of serious uncertainty? Do they have authority to make land use decisions and guide LUC at this scale and over long time horizons? Are they really interested in providing ecosystem services? Which services in particular (use the MEA classification). Have these models been used in real world decision making and by whom? At some point, the authors state: "Optimising models have the advantage of representing idealised conditions". Who decides what these ideal conditions are and for whom?

We have summarised these issues in Table 3 and in lines 201-210, and clarified the 'idealised' results (line 432). The models are primarily intended and used for exploratory modelling in academic research, with the IAP in particular having also been used for capacity building (with stakeholders and students).

SPECIFIC COMMENTS TITLE Two terms in the current title "How model paradigms affect our representation of future land-use change" may have to be modified. Model paradigms or modeling paradigms? Representation? The paper discusses estimates of future LUC, not representation which is something different and, in any event, it is reflected in the pre-defined land use classes and patterns used in the models.

Yes these are good points. We now use 'modelling paradigms' instead and changed 'representation', with a new title of 'How modelling paradigms affect simulated future land-use change'

"optimisation may be appropriate in scenarios that allow for coherent political and economic control of land systems, but not in scenarios where economic and other scenario conditions prevent the normal functioning of price signals and responses." This is correct: model results are valid if model assumptions hold... but caution is needed here. Reality may change and render model assumptions invalid...

Yes we entirely agree and have reworded to clarify our meaning (lines 22-25, 444-445).

What is the 'normal' functioning of price signals?

**This was poorly phrased. We now use 'equilibrium' instead.**

"structured comparisons of parallel, transparent but paradigmatically distinct models are an important method for better understanding the potential scope and uncertainties of future land use change" It is not clear that the present comparison is structured, or, at least, its presentation is not adequate and clear (see comments on conceptual framework and methodology). Parallel? Explain. Caution: understanding model results is not tantamount to understanding reality... Who wants to understand the potential scope and uncertainties of future LUC? (the issue of users mentioned above).

The structure of the comparison is now clarified as described above, as are the respects in which the models are 'parallel'. We agree that model results are distinct from reality, and in addition to the new text/details on model uses and users we now emphasise this point in lines 201-210.

"The optimisation model, in contrast, maintains food supply through intensification of agricultural production in the most profitable areas, sometimes at the expense of active management in large, contiguous parts of Europe."

This is unclear... active management? large, contiguous parts of Europe?

**Now rephrased (lines 21-22).**

**INTRODUCTION**

"Computational models of the land system are essential in supporting efforts to limit climate change and reverse biodiversity loss (Harrison et al. 2018; Rogelj et al. 2018)" These are not the only reasons for using these models. I suggest to broaden this sentence to encompass environmental and socio-economic change.

**Revised as suggested.**

"The need to radically alter human land use to avert social-ecological breakdowns".

This is unclear. Is it supposed to describe the aim of planned and/or unplanned land use change? If yes, the statement should be obviously modified.

Moreover, LUC modeling is used to analyse the impacts of past and current LUC, not only to project LUC under future scenarios.

This sentence refers simply to the unsustainable impacts of current land use. We have clarified this and now mention other model uses (lines 34-36).

"Because other methods are not available to generate alternative findings" Of course, there are other methods, both quantitative and qualitative (e.g. Delphi dating back to the 1970s), as well as mixed methods.

Yes this was poorly phrased and has now been altered accordingly (line 44).

"This could be particularly misleading in social systems such as those underpinning human land use, where no universal laws or predictable patterns exist to guide model development, and modellers must instead choose between a range of contested theoretical foundations, practical designs and evaluation strategies (Brown et al. 2016; Meyfroidt et al. 45 2018; Verburg et al. 2019)." This is a confusing sentence, especially the first part. It mixes up several notions and issues. It should be simplified and clarified.

Comment: predictable patterns are rare in nature also as complexity theory underlines and experience reveals.

We have split and revised this sentence (lines 46-49). We agree that predictability is rare in natural systems, but regard it as a particularly important issue in human/social systems where human behaviour introduces extra challenges beyond complexity.

"In this complex context, the proper analysis and interpretation of model outputs is just as important as proper model design"

Irrespective of context, the good modeling practice starts from the theory (however instrumental this may be) about the problem/situation modeled, that guides the development of the methodology, model design (model specification) and implementation (analysis), and, of course, the interpretation of model results.

**We agree, and have emphasised that the current study aims to improve interpretation in particular.**

"Steps such as standardised model descriptions, open access to model code, robust calibration and evaluation, benchmarking, uncertainty and sensitivity analyses are all necessary to ensure that model results are used appropriately (Baldos and Hertel 2013; Sohl and Claggett 2013)." What is the meaning of the word 'used' here? Model users are not model builders and vice versa. Clarify and modify. Moreover, the real test of a model's usefulness is model verification, not simply validation, however difficult to carry out (see, O' Sullivan).

We have changed 'used' to 'interpreted'. Model uses certainly extend beyond the uses model builders put them to, and accurate interpretation underpins them all. We also agree that model verification is crucial and involves more than the steps we mention here, and have amended the sentence to reflect this.

"However, while comparisons of model outputs have been made (Lawrence et al. 2016; Prestele et al. 2016; Alexander et al. 2017), their ability to link particular outputs to particular methodological choices has been limited."

This is absolutely reasonable because the factors affecting model performance and results are interdependent, important factors may be missing and/or intangible, data are unavailable or inadequate, contingencies modify system characteristics and relationships, etc.

There are certainly limits to linking model design to model outputs, but we would suggest that these limits have not yet been reached, partly because model comparisons have been relatively few and relatively limited in their scope, for instance not including detailed comparisons of quite distinct but parallel models (in the sense of sharing application coverage, resolution, contextual data etc.) of the kind that we make here. We now more clearly present this as the motivation for the study (lines 64-74, Section 2.3), and carefully define the basis of the comparison in terms of model similarities and differences as described above.

"Conceptual research suggests that large areas of system behaviour remain under-explored as a result (Brown et al. 2016; Huber et al. 2018; Meyfroidt et al. 2018), with the likely consequence that established findings have implicit biases and blind spots."

Yes, this is very true and should problematize efforts at comparing different models.

We have extended this point to acknowledge that comparisons can illuminate some but not all such biases and blind spots (line 73).

**METHODS**

This section should start with a description of the conceptual framework of the comparison.

Now added in Section 2.3 (we do not place this at the start of the Methods section as we feel the need to establish some of the model details first, but also explain the conceptual basis of the comparison in the new text in the Introduction and earlier in the Methods).

IMPRESSIONS IAP. "Within this cross-sectoral modelling chain, rural land use is allocated within each 30-year timeslice according to a constrained optimisation algorithm that maintains equilibrium between the supply and demand for food and (as a secondary objective) timber, through iterating agricultural commodity prices (cereals, oilseeds, vegetable protein, milk, meat etc.) to promote agricultural expansion or contraction (Audsley et al. 2015)."

So, IMPRESSIONS' aim is to optimized food supply? This should be mentioned from the outset.

We now clarify from the outset that the model aims to satisfy food demand (taking account of net imports), and does so optimally subject to constraints imposed by biophysical and socio-economic conditions (lines 118-119, Table 3).

"similar production conditions (based on soil and agroclimate)," Production conditions include many more factors, such as economic, fiscal, technological, institutional, etc.

Yes, we were referring more specifically to biophysical conditions here, and have amended as such ('similar biophysical conditions (based on soil and agroclimate)') (line 120, also 120-127).

"profitability thresholds used to determine which land use and management intensity is allocated to each cluster."

There are other factors (planning, policy, cultural...) that affect land use allocation and management intensity.

Moreover, caution is needed to interpret these findings given the spatio-temporal and conceptual (land use classes) aggregation of large-scale models. The 'managers' are not real people... so, what is the meaning of profitability? Profitable to whom?

We describe land allocation in both models more fully in the revision, and in particular the role of these and other additional factors (many of which affect allocation) – Tables 1-3 and lines 118-127, 157-158, 186-191, 201-218, 223-224, 345-346, 428-430, 439-442. Profitability is used here to mean

the simulated profit available for a particular level of production in a particular cell, and indeed does not refer to profit to real land managers, which we now emphasise.

**"Land use proportions within each 10' x 10' grid cell represent the aggregations of the solutions for each (up to 40) associated cluster."**

What is the meaning of so aggregate results?

We now clarify the derivation and interpretation of the aggregated results in Table 3 and lines 119-127. The aggregation is the result of spatial weighting of the optimised land use solution for each cluster containing the grid cell in question. The clustering recognises that different biophysical conditions (soil and agroclimate) differentially influence the suitability, productivity and profitability of different crops and different agricultural systems (arable, dairy etc.), leading to heterogeneity in agricultural land use within a grid cell.

"Modelled land manager agents compete for land on the basis of their abilities to produce a range of ecosystem services that society is assumed to require..."

How do you know that this is the aim of the land decision makers, especially at such coarse level of aggregation? I.e. to produce ecosystem services?

Psychological (emotional, political and institutional factors) regulate their relationships. Power relations are also important determinants of land managers behaviour (however coarse their representation is).

This text describes processes in the model, as distinct from reality – the competition for land on the basis of ecosystem service provision is a modelling assumption analogous to the allocation of land uses on the basis of profitability in the IAP. We describe the inclusion/exclusion of particular factors in more detail in the new model description table (Table 3) and also address missing factors and processes in the text as described above, as well as making it clear that we address differences between the models here and not differences between them and observed land use change.

"Land use productivities"

These should be defined taking into account the very high level of aggregation of the models. One question is: given the high level of aggregation, how much sense do they make as goals of land managers and decision makers?

Another question is: which factors influence these very aggregate productivities?

We have rephrased for clarity (this refers to the yields / ecosystem service provision levels of the different simulated land use systems (crops, grassland, forestry etc.) under the agronomic scenario conditions) (lines 152-153).

"In CRAFTY-EU, these services are crops, meat, timber, carbon sequestration, recreation and landscape diversity. We therefore also compare ecosystem service production levels, which account for exact forms of management simulated in each cell"

The ecosystem services should be first defined in terms of the 4 main groups defined in MEA (2005) and then shown how they are operationalized.

Moreover, the sentence should be edited (crops, etc. are not services...).

We have revised as suggested (lines 138-142).

"In this case, these functions are linear and equivalent for all services, meaning that the benefit of production of each service increases equally per unit of unmet demand." A very restrictive assumption indeed...

This is deliberately restrictive, yes. It allows us to compare an equal weighting of service provision (in CRAFTY) with a focus on food production (in the IAP), and avoids a more complex but equally arbitrary weighting that would make results harder to interpret. We now highlight this in line 145.

"Importantly for this study, CRAFTY-EU is parameterised on the basis of the IAP, taking IAP outputs as exogenous conditions and replacing only the land allocation component to provide alternative land use projections under identical driving conditions." This is unclear.

Comparison of two models when one takes input from then other?

We have now extensively clarified this point, presenting the relationship and inputs/outputs linking the two models in the new Fig. 1, and giving further description in Tables 2-3 and throughout the Methods section.

"For ecosystem services with economic values (meat, crops and timber), agents in CRAFTY therefore make production choices consistent with this basic level of economic rationality." First question: who are these agents at such a high level of aggregation and what is the meaning of

economic rationality in this case? Second question: what about non-economic benefits?

We have revised this sentence for clarity (lines 160-162). As described in the text, modelled agents do not correspond to real-world actors, but are used to capture elements of their behaviour within localised land systems. Non-economic benefits are also included in CRAFTY.

A note regarding NUTS2: They do not represent a uniform, EU-wide spatial division system and they differ significantly among countries.

We agree. There is no ideal resolution at which to make this comparison, but we chose NUTS2 as an established system to complement the results we present at cell and European scales.

**Subsection: 3.1. Aggregate comparison**

The presentation of the results is descriptive and technical.

The discussion is rather loose and tiresome with reference being made to the scenarios that have not been adequately described. The presentation of the results is rather boring and may not make a lot of sense to the reader.

We have added description of the scenarios and results in the revision in Table 1. We have also edited the text to ensure clarity and interest where we can.

One question here is: What is being interpreted? Model land use classes or real world land use? Also, the processes that produce LUC differ between countries and for each land use class, among other factors.

Model land use classes are being interpreted. We now clarify this prominently in describing the conceptual framework of the exercise (section 2.3). We agree that processes differ and have added discussion of this important point as described above.

"because of the gradual decision-making of agents" What does gradual decision making mean?

This refers to agents' decisions having some probability of being delayed across multiple timesteps (representing years), rather than taking immediate effect. We have explained this in lines 282 and Table 3.

"Conversely, CRAFTY responds most strongly to scenarios in which agricultural productivity decreases because its design emphasises changes in capitals that support production (climatic or socioeconomic), as is particularly clear in SSP3." The meaning is unclear.

Now rephrased (lines 287-294).

**Subsection: 3.2. Spatial comparison**

This is not spatial but 'geographical' comparison because it refers to geographic areas in Europe. The term 'spatial' is a general term and NOT identical to 'geographical'.

**We now label this as a geographical comparison.**

"In SSP4, the IAP projects substantially more very extensive and forest management than CRAFTY's more intensive results," extensive WHAT? more intensive WHAT?

**Now revised (line 300).**

Subsection 3.3. Convergence experiment The convergence/divergence of the results of different models owes to a host of factors, several of which were not examined here.

So, I wonder what is the meaning of carrying out this experiment?

We now clarify the purpose of this experiment in the text: it is indeed intended to identify these factors in this particular case (lines 178-180, 243-244, 320-324). The observed divergence in this scenario is partly due to conditions differing in the models (because food prices rise higher in the IAP and production levels fall lower in CRAFTY). By controlling these differences, we are able to identify additional factors that cause divergence – and in this case they reflect basic modelling assumptions, the effects of which would otherwise remain obscure.

**4. Discussion and conclusions**

I would have preferred a separate and proper Conclusions section.

The discussion of the results might be more meaningful to be combined and integrated with the presentation of results (previous section).

**We have written a stand-alone Conclusions section (section 5) but prefer to keep results and discussion separate as we find it important to establish technical findings before interpreting them.**

"Understanding the contributions of different modelling paradigms to land use projections is important for two main reasons. The first reason is that almost all large- to global-scale land system models share a single paradigm (economic optimisation of land uses). The second reason is that different paradigms are known to produce very different outcomes, but for reasons that remain unclear"

The question regarding those 'unclear reasons' is: what were the reasons hypothesized in this study? Otherwise, how was the comparison of model results carried out?

In this case, we hypothesise that the decision making / allocation paradigm is one dominant source of uncertainty in land use modelling, as opposed to uncertainty in crop yields, biophysical conditions etc. Hence we keep the latter factors common between the models and explore how different factors that influence decision making (profitability; demand; socio-economic conditions) affect the models. We set this hypothesis out in the revision as described above and use it to structure the methods and results section (in particular the new section 2.3).

"The focused comparison presented here is therefore intended to identify and explain key differences between models representing major, distinct paradigms to project land system dynamics on the basis of complex and integrated processes founded on a small number of key, transparent assumptions".

What are these key, transparent assumptions? Were these key differences explained? The issue of the conceptual/theoretical framework underpinning the comparison is critical here.

**We elucidate the underlying framework in the revision with particular emphasis on these key assumptions that differ between the models (section 2.3 and Table 3).**

"An overarching distinction is apparent between the basic assumptions underlying the models. The IAP is an example of a 'topdown' model that simulates change at the system-level – in this case through an assumption of constrained economic optimisation - while CRAFTY is an example of a 'bottom-up' model that simulates change at the level of individual decisionmakers – in this case through an assumption of behavioural choices made at the level of local land systems (Brown et al. 2016).

This basic difference affects the rate, extent and pattern of simulated land use change. The consequences of top-down and bottom-up perspectives is apparent in the main forms of land use change as the models respond to scenario conditions.... "

I am not sure if any adequate explanation of the implications of different model assumptions is offered in the above excerpt.

**Explanation of implications is in the text that follows the quoted sentences. We believe the additional structured comparison of the models in the revision helps to explain the assumptions and their implications.**

"This difference is also apparent in our convergence experiment, where increased imports in the IAP lead to reduced agricultural area,"

But, doesn't it happen the other way around? Land is abandoned, then production drops and necessitates increase in imports... Do I miss something?

We now clarify this in lines 320-324 and Fig. 5. The convergence experiment involved pre-emptively increasing imports in the IAP to mimic the lower European production levels generated by CRAFTY.

"One consequence of simulating demand and supply of a range of ecosystem services is that the relative economic support available for food production becomes a key determinant of the balance of different land uses"

What is the theoretical explanation offered?

The case may be that, because agriculture is the most extensive land use and occupies a larger number of cells, it leads to the results obtained. In other words, the results may owe to technicalities and not to real world market and social behaviour and responses.

An interesting point, and one we address in the revision in lines 391-393. The result is certainly due to technicalities, in the sense that the model is sensitive to the relative valuation because that is the basis for simulated land competition, but reflects the reality that land use, as primarily economicallydriven, is subject to the relative economic support for food production and for other ecosystem service provision.

"In both models, the simulation of the European land system as distinct from the rest of the world requires implicit assumptions about conditions in other regions and their relationships to Europe. As conceptual alternatives, therefore, neither of these necessarily capture the true dynamics of food prices and production levels, which remains a major challenge for land system modelling (Pedde et al. 2019; Müller et al. 2020)."

This is a correct remark. The exogenous factors have been incompletely modeled. Their inclusion may have further differentiated the results of the two models.

Yes, we agree. We have added a sentence to emphasise that different approaches to modelling exogenous factors would likely introduce even greater differences (lines 403-404).

"Cell level decisions" Do cells decide? ( ). Which theory concerns cells?

**We have rephrased as 'simulated decisions affecting individual cells' (line 410).**

"Indeed, their primary strength may be their ability to use theory as a guide to processes and conditions that empirical data and optimising models do not cover (Gostoli and Silverman 2020). " This is partly true re ABMs. The question is: which theory do they use?

We have added a comment emphasising the choice of theory (line 438), although we see this as a distinct issue to the inclusion of a wider range of theories as an end in itself.

"The greatest value of these two approaches may therefore lie in their ability to provide alternatives; a value that is realised only in the (currently rare) cases when model assumptions are clearly communicated and when analogous models such as those used here are available for comparison (Polhill and Gotts 2009; Müller et al. 2014; Rosa et al. 2014)." This sentence is unclear.

**Now rephrased (lines 444-448).**

THE LAST PARAGRAPH of the paper is a rather unstructured list of open issues and future research directions that does not flow directly from the preceding analysis and does not offer much direction centered around a concrete model aim...

This paragraph has now been edited and is followed by a new conclusions section.

The question is: why is it necessary to keep these modeling paradigms when alternatives are already tested and more meaningful? E.g. multi-paradigm modeling.

We are not entirely sure in what sense 'multi-paradigm modeling' is being used here, but the models and paradigms represented here have also been tested and found to be meaningful in a number of ways (and are of course widely used). In any case, to the extent that different paradigms are present within 'multi-paradigm modeling', we see our basic premise of understanding how underlying assumptions influence model outputs as still relevant.

It might be useful to discuss in the conclusions, issues of model users, use and usefulness that might further justify pertinent future research

Yes, we find this a good suggestion and added some discussion of models users/uses as described above.

**TECHNICAL COMMENTS**

Table headings should be placed at the top of the Table. P. 23 The heading of the Table is long ... it should be much shorter! ecosystem service supply ... SERVICES

We have placed Figure and Table headings according to the journal's template, and retain the information in the heading of Table A1 as we find it important for interpretation. Ecosystem services has been changed.

**Reviewer 2:**

Overall this is a good paper with some really interesting results. With some additional improvements to the figures/analysis it could be excellent and make an excellent contribution.

Thanks for the positive comments and the suggestions.

I am mostly viewing this as a scholar who uses land use projections, and the discussion of the different approaches and how they differ is really important. I like the introduction in general but a bit more detail would be appreciated.

**We have now added more detail, in common with the responses to R1.**

I would also like the authors to discuss how observational data is incorperated into this models, or not. The usual standard for earth system science, is a lot of comparison to observations, so please explain how each of these paradigms try to make sure they actually compare well to observations, especially of historical land use change trajectories, or if they do not do such comparisons. If currently there is no comparison, this could be a way to differentiate these different approaches to see which is more accurate.

Thanks, an important point. We now detail specific and general use of observational data in the revision, particularly in the model description/comparison table (Table 3) and in lines 420-425, with the new Section 2.3 also being relevant in places. In general, optimising models have indeed been more often compared against data but we provide specific details for these models.

I also think a bit more analysis could be helpful in the figures to synthesize a bit more. Details below.

Figure 2a: the dark (IAP) vs. lighter (CRAFTY) symbol isn't clear enough here: I recommend you make more of a matrix with left being light colors and right being dark colors and showing then that the right ones are IAP and left ones are CRAFTY. I stared at the plot for awhile before I understood what the dark grey and light grey was in the figure.

Thanks for the suggestion; we have taken this approach.

Figure 2b: the colors aren't really different enough here, and the same issue with the dark vs. light colors.

**The colours have been changed.**

Figure 2 in general: Would a difference plot work better for this? Or a % difference? There are so many similar bar graphs?

We were unsure how well this worked, so have retained the bar charts. While there's a lot of information in them we think it's probably easier to extract all the information in this format, especially after the colour and legend changes.

Figure 3: white means two things here: not included, and not land? Please try use grey for one of those so this is clearer. Maybe you want difference plots here instead of these contrasting, but similar plots? Are there patterns of where in particular the differences are important that you can find and call out?

We have changed the white 'not included' (no difference) to grey. We also agree there was too much here, so have made a new replacement plot showing the total differences between the models across all scenarios (Fig. 4). This highlights the most persistent and informative differences, and we retain the full scenario plots in the new Appendix B for information.

Figure 4: this graph is not self standing enough: describe what is on the left versus the right, why the arrow goes in the opposite direction on the bottom, everything needs to be explained. Describe the alternative parameterizations briefly here in the figure

**caption.**

**We have amended this figure as suggested, thanks.**

**Reviewer 3 (Robert Huber):**

I think this is a valuable contribution to the discussion on how computational models can inform political and social efforts toward more sustainable land systems on a large spatial scale. I find it very important to explicitly discuss underlying paradigms in models of land-use change and I think this contribution is an important step to improve our understanding of how the paradigm affects the interpretability and validity of such models. In my view, however, the current version of the manuscript could be improved by describing the model paradigms more explicit and by a more careful presentation of the input-output relations in the result Section. In addition, I think the contribution would gain from discussing the implications of the different paradigms to inform what the authors call "efforts to limit climate change and reverse biodiversity loss". I would like to specify my general comment below.

**Many thanks for the positive feedbacks and useful suggestions.**

1) I think it would be important to introduce the two model paradigms earlier in the manuscript. In my understanding, the first part of the discussion (lines 239-260) is something that defines the research design and should not appear as something the authors would like to discuss. In addition, I think it would help the reader if the authors would also discuss and classify/categorize these two paradigms a bit broader e.g. in the context of their own work on ABMs and their theoretical and philosophical background (Arneth et al., 2014; Brown et al., 2016).

**We followed this suggestion, moving the text referred to back in the manuscript, detailing the paradigms more carefully in the new model descriptions (including the new Table 3) and relating them to earlier publications as suggested (lines 75-77 and Section 2.3).**

2) In the same vein, I think that the discussion of the paradigm should also include implications from the different mathematical model implementations. If I understood the models correctly, IAP maintains equilibrium between the supply and demand for food while agents in CRAFTY compete for land-uses having a satisficing behavior including non-economic benefits. The point I'd like to make is that models based on rational economic behavior usually are characterized by switching from one corner of the mathematical solution space to another. I do not know whether this describes IAP adequately. However, the outputs seem to suggest that CRAFTY results are always more balanced than the economically driven IAP results. Thus, I would suspect that IAP jumps to corner solutions. If this is true, then this would also be known before the comparison. There might be other direct implications of the mathematical implementation of the models for the interpretation of the output. This could be introduced and discussed in the context of model paradigms.

A good point, and we highlight issues of this sort that are clear prior to the comparison in lines 211-218, 372-379 and Table 3). This attribution is slightly complex – for instance the IAP simulates and aggregates up to 40 clusters in each grid cell that produce different solutions, with changes also possible within a land use class (e.g. different crop selections) (as now clarified in the text), but there are certainly elements of model design that mathematically pre-define model outcomes.

3) I also found it difficult to follow the input-output description in the text but also the figures. The authors use the pre-defined abbreviations for the different climate and socio-economic scenarios. I understand that there are reasons not to give explicit names to these scenarios. Nevertheless, it

makes the presentation of the comparison in this contribution very demanding. As a reader unfamiliar with the exact definition of each of the socio-economic scenarios, I always had to cross check what SSP3 or SSP1 now exactly implies with respect to the input assumptions. Since there is no description of the socio-economic scenarios in Section 2.2, I had to use O'Neill et al. to be able to follow the result Section. In addition, I did not really understand how the convergence scenario was developed. I think the manuscript would profit from a concise description of the socio-economic scenarios and how these scenarios affect the underlying assumption in the model exercise e.g. production functions, demand levels etc. This would help the reader to connect input- and outputs in the different models. Personally, I would find it also helpful if there would not be abbreviations for the socio-economic scenarios. This could make it easier and more accessible to the reader e.g. in Figure 1.

Thanks for highlighting this. We have added a scenario description and implementation table (Table 1), and also use the acronyms alongside more meaningful names on most occasions. We've also provided substantially more detail about how the models represent the scenarios (Tables 2 and 3, Fig. 1) and fuller description of the convergence experiment (320-325, Fig. 5).

4) In this context, I also had the impression that the authors did not adequately address and discuss the uncertainty with respect to model inputs. For example, the author writes that there is a "greatly increased productivity" in the scenario RCP8.5-SSP5 and consequently, the IAP model suggests that the supply of crops and meat can increase more than 30% with less than a third of the area of intensive agriculture (comparing Figure 1 and Figure 2a). The increase in productivity, in contrast, did obviously not affect land-use in CRAFTY. However, I would expect that a change in the productivity increase would considerably lower the extreme solution in the SSP5 scenario. Maybe that is something the authors wanted to address with the "convergence" comparison: Look at the sensitivity to input parameters of specific importance. I think this would deserve more attention. Maybe the authors can include more than just two input variations (increase in imports and food values) and discuss the results in the context of input uncertainty that seems to have very different impacts in the two model paradigms.

An interesting point, and it's quite correct that we didn't deal with uncertainty/sensitivity in any depth. To rectify this we give details of uncertainties and sensitivities in the model descriptions and in the results & discussion (Table 3, Section 2.3, lines 372-380, 474-480). While we're wary of adding more experiments here in addition to substantial extra explanation as suggested by the reviewers, both models have been quite extensively assessed in sensitivity and uncertainty analyses in the past, including with respect to scenario conditions, and we therefore hope that the summaries of these findings we now provide improve interpretation of the findings.

5) With respect to the methods, I acknowledge that these are well documented and state-of-art models that are suitable for comparing the effect of different model paradigms on future land-use change. However, one sentence in the manuscript confused me. The authors write (lines 113ff): "CRAFTY-EU is parameterised on the basis of the IAP, taking IAP outputs as exogenous conditions and replacing only the land allocation component to provide alternative land use projections under identical driving conditions." What is implied here by taking the output of IAP as exogenous conditions for CRAFTY. It would not make sense to use outputs as inputs in another model and then conclude that the models have different outputs. I'm sure this is a misunderstanding (culpa mea). However, the authors should be clearer in what they do here. What are these conditions (except land-use) and how do they affect the comparison? Maybe the solution here goes hand in hand with the reply to my comment 3. However, I would suggest that the authors explain the data exchange between the models in more detail.

Yes agreed, and we now explain this properly in the revision, including via a model diagram (Fig. 1) that shows the relationship of the two models and their input/output sharing. We also give extra details about model inputs and outputs in Tables 2 and 3, as well as at several points in the text.

6)The last comment I'd like to make is probably also the most difficult to address. When looking at the results, I had the impression that the two model paradigms lead to really large differences despite using the same scenarios (e.g. in Figure 3). The authors also state that the "greatest value of these two approaches may therefore lie in their ability to provide alternatives". But if these models should inform "efforts to limit climate change and reverse biodiversity loss" what do these alternatives imply? Obviously one would come to very different conclusion what to do concerning e.g. biodiversity loss depending on the model paradigm (irrespectively of the scenario). Given the potentially contradicting (policy) conclusions from these "alternatives", critics of mathematical modelling could argue that this "invalidates" such simulations. One can get any result by choosing the "right" paradigm. I'm aware that the contribution does not attempt to address all of the caveats in model design, analysis and interpretation mentioned in the Introduction. However, I had the impression that the authors take refuge in discussing "technical integration" of models. But how could such a hybrid modelling approach solve potential contradictions? In climate change modeling, model ensembles are a way of addressing different underlying functionalities of models. However, it seems to be impossible when looking at the results of this exercise. I think this point should at least be discussed: what if model paradigms prevent instead of foster discussions on how to use modelling of more sustainable land systems on a large spatial scale? I have the impression that the authors should also discuss the value of theoretical underpinnings and conceptual frameworks (which may be more important in this context) than just "more data from another discipline on another spatial level" (which is my simplified interpretation of the last paragraph).

We find these excellent suggestions and fair criticisms. It is probably true that we take refuge in technical issues to some extent! This is partly because we wish to establish basic differences here, but we should have better addressed this overarching issue. We have therefore added text to link our findings to the motivating question of model uses (including in Table 3, and lines 201-210), and return to this issue in particular in the new Conclusions section (section 5).

**Minor comments**

**What is the unit of the Y-axes in Figure 1?**

**Now defined in the figure legend.**

I would prefer if the difference between IAP/CRAFTY in the figures would not be represented by the level of shading only. Maybe the authors can use a different pattern or something that makes it easier to distinguish the models.

**We agree this was confusing. We've settled on the suggestion of Reviewer 2 of changing some colours and providing a more informative legend in this revision.**

I found the caption in the Figures not self-explaining (and I have to say a bit cryptic in the beginning). I do not really understand why some specific information is given in one Figure but not in the other. I think that the authors should try to make the caption self-explaining (in a way).

**We've tried to standardise and complete all figure captions and hope they are now self-explanatory.**

On line 302ff, the authors state that "Conversely, (constrained) optimising models like the IAP produce idealized results that (...) can use flexible spatial dependencies as proxies for processes such as imitation, diffusion of knowledge or the formation of social norms (). Are you sure that

knowledge diffusion and social norms fit into the economic framework of IAP? Not sure I understood this sentence.

It's true that these processes do not really fit, but they can be accounted for to some extent through the use of proxies; especially undefined spatial correlations in land use changes. We have now clarified this point in lines 413-417.

**How modelling paradigms affect our representationsimulated of future land-use change**

Calum Brown1, Ian Holman2, Mark Rounsevell1,3

[revised manuscript text omitted]

680 Table 3: Summary comparison of the two models used in this study across a range of characteristics, many of which stem from the distinct modelling paradigms used. Further details are provided in the text, and references cited there.

| Land use classes for comparison | Explanation                                                                           |
|---------------------------------|---------------------------------------------------------------------------------------|
| Intensive agriculture           | Intensive forms of agriculture primarily dedicated to crop production but including   |
|                                 | some grassland                                                                        |
| Extensive agriculture           | Extensive forms of arable and pastoral agriculture                                    |
| Pastoral agriculture            | Dedicated and primarily intensive pastoral agriculture                                |
| Very extensive management       | Management for any service that is of the lowest intensity and leaves land in a near- |
|                                 | natural state                                                                         |
| Forestry                        | Active management for timber extraction and other forest services                     |
| Other/no management             | Land that is not actively managed for agriculture or forestry, but which can have a   |
| _                               | range of natural or human-impacted land covers                                        |

Table 41: Land use classes used in the comparison and their composition. Derivations from the full range of CRAFTY and IAP classes are given in Table A1.

---

## Referee Report (RR1)

**How modelling paradigms affect simulated future land-use change**

GENERAL REMARKS

The revised paper has satisfactorily and competently addressed most of the comments I have provided on the original paper.

The merit of the paper is twofold. First, it raises awareness regarding the importance of modeling paradigms and other model assumptions in determining the results the models produce, necessitating their careful and situated interpretation and cautious use in making policy decisions. Second, by employing a systematic and focused comparison, it offers a comparative assessment of a top-down/aggregate with a bottom up/disaggregate model and discusses their relative worth and role in making decisions, despite the several open issues remaining for each modeling paradigm.

Moreover, the presentation and discussion of the models reveals the sensitivity of their results to the scenarios used and inputs from other models (see, e.g. section 2.2).

The authors are careful to underline the coarse level at which models operate and limits their application to offering broad indications of potential land use change under several socio-economic and climatic scenarios. They also discuss several open issues that may be addressed, but also may not be addressed, in the context of these modeling exercises.

A class of exogenous forces affecting land use change and may be probably tested at a coarse level concern disruptions owing to pandemics and major political events. These cause perturbations in labor supply, movement of people and products, tourism, etc. and may precipitate land use changes at a not-so-far future. It might be interesting to compare the aggregate and the disaggregate model predictions under these conditions to help further evaluate their overall significance.

Selected, mostly editorial, comments are offered below.

ABSTRACT

11-13

"In this study, we compare two pan-European land use models that are based on the same integrated modelling framework and utilise the same climatic and socio-economic scenarios, but which adopt fundamentally different modelling paradigms."

It may be written more simply to avoid confusion:

"In this study, we compare two pan-European integrated land use models, that utilise the same climatic and socio-economic scenarios, but which adopt fundamentally different modelling paradigms."

INTRODUCTION

41

"to guide models' representation of human behavior"

"to guide the representation of human behavior in models" reads better.

58-59

"These previous comparisons reveal a major challenge: the shortage of models that take distinct approaches at similar geographical and thematic scales"

What is a thematic scale?

Did you want to write:
"These previous comparisons reveal a major challenge: the shortage of models that take distinct approaches at similar geographical scales and thematic areas"?

63

"Conceptual research suggests that…"

What is conceptual research?

69-71

"We use the term 'modelling paradigm' here to refer to a coherent methodological and theoretical approach, and specifically the 'top-down' and 'bottom-70 up' approaches frequently identified as paradigms in the literature (Brown, Brown, & Rounsevell, 2016; Couclelis, 2002)."

This sentence is unclear and confusing. Edit.

193-194

"We therefore also compare ecosystem service production levels, which account for exact forms of management simulated in each cell."

How exact can be a simulation of a form of management in a 16kmX16km cell?

TERMINOLOGY

I suggest that instead of using the term 'geographical' to distinguish spatially disaggregate (spatially explicit) from aggregate analysis, the term 'territorial' is more suitable and supported by the fact that the spatially disaggregate estimates are using the NUTS classification scheme which is a territorial and not geographical scheme mainly.

In the same spirit, I suggest replacing the term 'overall land use change' with 'aggregate land use change'.

The usage of these two terms – aggregate and spatially disaggregate (or, spatially explicit) – can be explained from the beginning and used consistently throughout the paper.

See some examples below.

3.1 Overall EU-level comparison

You can write it as: EU-level aggregate comparison

3.2 Geographical comparison

You can write it as: Territorial comparison

258

"Within the overall differences between model results exist some consistent spatial and geographical patterns (Fig. 4)."

I suggest replacing 'geographical' with 'territorial'

Fig 4: Geographical differences … I suggest using 'territorial' instead of 'geographical'

311-312

"The consequences of top-down and bottom-up perspectives are apparent in the forms, extents, rates and patterns of land use change as the models respond to scenario conditions."

Replace plural with singular number (form, extent, rate):

"The consequences of top-down and bottom-up perspectives are apparent in the form, extent, rate and patterns of land use change as the models respond to scenario conditions."

343

"All of the models' results" reads better as "All the results of the models"

404

"we can only make tentative conclusions", rewrite "we can only draw tentative conclusions"

Table captions should be placed on the top (not bottom) of the Tables

Figure 1: Simplified schematic showing … replace with

Figure 1: Simplified schema showing …

Figure 3a: Supply levels of services that both models attempt to satisfy demand for, in each scenario. Edit

Similarly edit Figure 3b.

Appendix B: Full geographical scenario results

I suggest instead:

Appendix B: Complete territorial scenario results

---

## Author Response (AR2)

**Reviewer 1**

GENERAL REMARKS

The revised paper has satisfactorily and competently addressed most of the comments I have provided on the original paper.

The merit of the paper is twofold. First, it raises awareness regarding the importance of modeling paradigms and other model assumptions in determining the results the models produce, necessitating their careful and situated interpretation and cautious use in making policy decisions.

Second, by employing a systematic and focused comparison, it offers a comparative assessment of a top-down/aggregate with a bottom up/disaggregate model and discusses their relative worth and role in making decisions, despite the several open issues remaining for each modeling paradigm.

Moreover, the presentation and discussion of the models reveals the sensitivity of their results to the scenarios used and inputs from other models (see, e.g. section 2.2). The authors are careful to underline the coarse level at which models operate and limits their application to offering broad indications of potential land use change under several socio-economic and climatic scenarios. They also discuss several open issues that may be addressed, but also may not be addressed, in the context of these modeling exercises. A class of exogenous forces affecting land use change and may be probably tested at a coarse level concern disruptions owing to pandemics and major political events. These cause perturbations in labor supply, movement of people and products, tourism, etc. and may precipitate land use changes at a not-so-far future. It might be interesting to compare the aggregate and the disaggregate model predictions under these conditions to help further evaluate their overall significance.

Many thanks for the comments and suggestion about applications to coarse level disruptions, which we now mention in the conclusions section.

Selected, mostly editorial, comments are offered below.

ABSTRACT 11-13 "In this study, we compare two pan-European land use models that are based on the same integrated modelling framework and utilise the same climatic and socio-economic scenarios, but which adopt fundamentally different modelling paradigms." It may be written more simply to avoid confusion: "In this study, we compare two pan-European integrated land use models, that utilise the same climatic and socio-economic scenarios, but which adopt fundamentally different modelling paradigms."

Changed as suggested.

INTRODUCTION 41 "to guide models' representation of human behavior" "to guide the representation of human behavior in models" reads better.

Changed as suggested.

58-59 "These previous comparisons reveal a major challenge: the shortage of models that take distinct approaches at similar geographical and thematic scales" What is a thematic scale? Did you want to write: "These previous comparisons reveal a major challenge: the shortage of models that take distinct approaches at similar geographical scales and thematic areas"?

Changed to "These previous comparisons reveal a major challenge: the shortage of models that take distinct approaches in similar geographical and thematic areas"

63 "Conceptual research suggests that…" What is conceptual research?

This is probably clearer without the reference to conceptual research, so we have now deleted it.

69-71 "We use the term 'modelling paradigm' here to refer to a coherent methodological and theoretical approach, and specifically the 'top-down' and 'bottom-up' approaches frequently identified as paradigms in the literature (Brown, Brown, & Rounsevell, 2016; Couclelis, 2002)." This sentence is unclear and confusing. Edit.

We have edited this sentence to read: "We use the term 'modelling paradigm' here to refer to a methodological approach that is based on a distinct theoretical description of the system in question; in this case 'top-down' and 'bottom-up' approaches frequently identified as paradigms in the literature (Brown, Brown, & Rounsevell, 2016; Couclelis, 2002)"

193-194 "We therefore also compare ecosystem service production levels, which account for exact forms of management simulated in each cell." How exact can be a simulation of a form of management in a 16kmX16km cell?

Yes, the word exact was misleading; we have now changed it to "actual", meaning the forms simulated in that cell rather than the generic label given to them.

TERMINOLOGY I suggest that instead of using the term 'geographical' to distinguish spatially disaggregate (spatially explicit) from aggregate analysis, the term 'territorial' is more suitable and supported by the fact that the spatially disaggregate estimates are using the NUTS classification scheme which is a territorial and not geographical scheme mainly. In the same spirit, I suggest replacing the term 'overall land use change' with 'aggregate land use change'. The usage of these two terms – aggregate and spatially disaggregate (or, spatially explicit) – can be explained from the beginning and used consistently throughout the paper. See some examples below.

Thanks for the suggestions, and we have adopted both terminology changes.

3.1 Overall EU-level comparison You can write it as: EU-level aggregate comparison

Changed as suggested

3.2 Geographical comparison You can write it as: Territorial comparison

Changed as suggested

258 "Within the overall differences between model results exist some consistent spatial and geographical patterns (Fig. 4)." I suggest replacing 'geographical' with 'territorial'

Changed as suggested

Fig 4: Geographical differences … I suggest using 'territorial' instead of 'geographical'

Changed as suggested

311-312 "The consequences of top-down and bottom-up perspectives are apparent in the forms, extents, rates and patterns of land use change as the models respond to scenario conditions." Replace plural with singular number (form, extent, rate): "The consequences of top-down and bottom-up perspectives are apparent in the form, extent, rate and patterns of land use change as the models respond to scenario conditions."

Changed as suggested

343 "All of the models' results" reads better as "All the results of the models"

Changed as suggested

404 "we can only make tentative conclusions", rewrite "we can only draw tentative conclusions"

Changed as suggested

Table captions should be placed on the top (not bottom) of the Tables

These were laid out following the journal's template so we have left them where they are pending any further instructions.

Figure 1: Simplified schematic showing … replace with Figure 1: Simplified schema showing …

Changed as suggested

Figure 3a: Supply levels of services that both models attempt to satisfy demand for, in each scenario. Edit

Edited to read: "Supply levels of services that both models attempt to satisfy demand for. Supply levels are shown for each scenario, and demand levels (derived from the IAP) are indicated by a red line for each service."

Similarly edit Figure 3b.

Edited to read "Supply levels of services that only CRAFTY attempts to satisfy demands for (while the IAP does not)."

Appendix B: Full geographical scenario results I suggest instead: Appendix B: Complete territorial scenario results.

Changed as suggested.

**Reviewer 2:**

The authors have done a good job responding to the reviews and improving the manuscript. My only comment is that in the abstract you should be a bit clearer that your result is that the answer is EXTREMELY sensitive to the approach, perhaps try to be quantative about how different the amount (for example) of intensively managed land is for several scenarios. This answer is clear from the graphs, but I don't see these numbers in the text, so they should be added to the text, conclusions and abstract.

Thanks for the suggestion, which we have followed in the abstract, results and conclusions.